# A density functional theory for ecology across scales

Martin-I. Trappe [1,2] ✉ & Ryan A. Chisholm [1] ✉

Ecology lacks a holistic approach that can model phenomena across temporal and spatial scales, largely because of the challenges in modelling systems with a large number of interacting constituents. This hampers our understanding of complex ecosystems and the impact that human interventions (e.g., deforestation, wildlife harvesting and climate change) have on them. Here we use density functional theory, a computational method for many-body problems in physics, to develop a computational framework for ecosystem modelling. Our methods accurately fit experimental and synthetic data of interacting multi-species communities across spatial scales and can project to unseen data. As the key concept we establish and validate a cost function that encodes the trade-offs between the various ecosystem components. We show how this single general modelling framework delivers predictions on par with established, but specialised, approaches for systems from predatory microbes to territorial flies to tropical tree communities. Our density functional framework thus provides a promising avenue for advancing our understanding of ecological systems.

Ecology has detailed mechanistic theories that capture the behaviour of a small number of interacting species at fine scales[1–3]. It also has broad 'unified' theories that ignore the details of individual species and their interactions, but can reproduce large-scale patterns, such as the relationship between number of species and area[4–7]. Nevertheless, theories bridging different scales are lacking. Ideally, models would be parameterisable at one scale of ecological organisation (e.g., the individual organism, or the species) and make accurate predictions at other scales (e.g., the local community or landscape). Meeting this challenge would allow knowledge from field data and limited experimental setups to be leveraged to predict the behaviour of more-complex, realistic ecosystems[4,8–12]. Such an enterprise calls for a diverse range of quantitative approaches.

Here, we draw on density functional theory (DFT)[13–16], which has been successfully applied to a staggering number of many-body problems in physics, involving scales from atoms to molecules to planetary cores[17–20]. DFT is based on the mathematical proof that all properties of all quantum systems can, in principle, be obtained from the density distributions of its constituents. At the core of DFT lies an

energy (cost) functional in terms of these density distributions. Of fundamental interest in physics are the density distributions with lowest energy (ground states and equilibria), obtained from minimising the energy over all permissible spatially varying densities. DFT predicts key properties of physical systems while automatically accounting for trade-offs between system components and constraints with efficiency and accuracy. The conceptual framework of DFT is the mathematically rigorous generalisation of the Thomas–Fermi (TF) model[21,22] from the advent of quantum mechanics. The development and fruitful application of DFT has sky-rocketed since then, with nowadays tens of thousands of yearly publications[23]. In light of the success of DFT in simulating many-body systems within a general framework, it is worth considering whether complex systems in the life sciences can also be modelled with DFT techniques.

While there are numerous DFT applications in molecular biology[24], only one DFT-inspired study has so far emerged in macrobiology: The authors of Ref. [25] inferred spatial density distributions of *Drosophila melanogaster* (fruit flies) in an experimental arena by pairing the grand-canonical ensemble of statistical physics with an energy

[1]Department of Biological Sciences, National University of Singapore, 16 Science Drive 4, Singapore 117558, Singapore. [2]Centre for Quantum Technologies, National University of Singapore, 3 Science Drive 2, Singapore 117543, Singapore. ✉e-mail: martin.trappe@quantumlah.org; chisholm@nus.edu.sg

functional. This statistical method accurately predicts the distributions of fruit flies in new environments. Here we develop a more general approach, which we term density functional theory for ecology (DFTe). The core of DFTe is an energy functional for ecosystems that merges ecological principles with insights from DFT. Though DFTe draws inspiration in particular from analogies with the TF model of multi-component quantum gases, it does not of course introduce any quantum effects into ecology itself. Our DFTe bridges ecological scales by allowing sub-components of an ecological model to be para-meterised separately and then combined to predict the behaviour of more-complex systems. We first describe the general approach and then present applications to a variety of ecological systems with different properties.

## Results

### The cost function as the key concept of DFTe

Many key questions in ecology, such as abundance redistribution in response to environmental change, ask for constrained equilibria—a goal shared by DFT in physics. At the core of DFT lies a cost function, the density functional of the energy $E$. Once the building blocks of $E$ are specified, its minimisation delivers the position-($\mathbf{r}$-)dependent equilibrium (viz., ground-state) densities $\mathbf{n}(\mathbf{r}) = \{n_1(\mathbf{r}), n_2(\mathbf{r}), ..., n_S(\mathbf{r})\}$ for the abundances $\mathbf{N} = \{N_1, N_2, ..., N_S\}$ of any interacting many-body system of $S$ species, such as a gas of several atomic species.

We have transferred the DFT narrative and methodology to ecology by building a cost function, the energy functional

$$E[\mathbf{n}] = E_{\text{dis}}[\mathbf{n}] + E_{\text{env}}[\mathbf{n}] + E_{\text{int}}[\mathbf{n}], \qquad (1)$$

whose minimisation balances species-specific costs due to dispersal, environment, and interactions, including resource consumption (the three terms on the right-hand side of Eq. (1); see Methods). When using square brackets in $f[\mathbf{n}]$, we declare that a function $f$ is a functional of the function $\mathbf{n}$. In referring to $E$ as an energy for a given realisation ($\mathbf{n}$, $\mathbf{N}$) of an ecological system, we quantify a relation to other configurations ($\mathbf{n}'$, $\mathbf{N}'$). The insight that a system aims at occupying the state of lowest (ground-state) energy among all possible system configurations is one of the fundamental principles of physics. Here, we construct the DFTe energy $E(\mathbf{N})$ such that its global minimum realises the ecosystem equilibrium with abundances $\mathbf{N} = \hat{\mathbf{N}}$. While we focussed on establishing the equilibrium properties of DFTe as a necessary step towards the modelling of dynamics, we note that $E$ represents an $S$-dimensional (hyper-)surface whose shape can determine (i) steady-state dynamics (see Methods)—in analogy to, for example, Kepler orbits in a gravitational field—and (ii) nonequilibrium dynamics, where also drivers such as stochastic environments can prevent equilibration.

The energy of a system is a measure of its ability to invoke change in another system. Twice the energy, relative to the other system, doubles this ability—energy is additive. But this does not imply that the energy of a system decomposes into a sum as in Eq. (1). We base the additive decomposition of $E$ on our understanding of how constituents of physical systems come together to assemble the energy and determine the functional form of $E$ by introducing and building on intuitive concepts in analogy to the energy components of physical systems. For example, for the dispersal energy $E_{\text{dis}}$ we adopt the TF kinetic energy expression for species $s$:

$$E_{\text{dis}}[n_s] = \frac{\tau_s}{2} \int_A (\text{d}\mathbf{r}) \, n_s(\mathbf{r})^2, \qquad (2)$$

where $\tau_s$ is a species-specific dispersal pressure constant, and the integral is over all spatial positions $\mathbf{r}$ in a focal area $A$. Equation (2) encodes an intra-specific pressure of species $s$ and implements one form of dispersal where, for example, individuals repel conspecifics from territory; note that positive (negative) energy density encodes

repulsion (attraction). The functional form $E_{\text{dis}}$ in Eq. (2) implies a conspecific negative density dependence, relative to other energy components whose integrands scale less than quadratically with density. We emphasise that we neither aim at describing all forms of dispersal via Eq. (2) nor necessarily need to include $E_{\text{dis}}$ even if a specific ecosystem exhibits dispersal, which can potentially be modelled with other energy components. The functional in Eq. (2) is species-specific due to $\tau_s$, and we assume that the quadratic density dependence is the result of an individual-level mechanism that is not necessarily known to us. The second term in Eq. (1) is the environmental energy, which for each species $s$ we model via analogy to the external energy in physics:

$$E_{\text{env}}[n_s] = \int_A (\text{d}\mathbf{r}) \, V_s^{\text{env}}(\mathbf{r}) \, n_s(\mathbf{r}), \qquad (3)$$

where $V_s^{\text{env}}(\mathbf{r})$ governs the effect of the environment on the energy for species $s$. Equation (3) parallels the parameterisation in ref. [25] and models, for example, habitat preference and external influences, such as spatially varying climate and local deforestation stress. We may illustrate the effect of Eq. (3) for an amphibian species $s$ by assigning values $V_s^{\text{env}} < 0$ in a lake, while $V_s^{\text{env}} > 0$ in the surrounding desert penalises high densities $n_s$ as we minimise the total energy that includes $E_{\text{env}}[n_s]$. Summing Eqs. (2) and (3) over $s$, we obtain the respective total energy components displayed in Eq. (1).

We emphasise that we do not pursue the hopeless goal of establishing DFTe based on actual chemical quantities and mechanisms. Rather, we use our DFTe and its energy functional $E$, derived by analogy with DFT and informed by ecological principles, as a tool for parameterising sub-components of ecological systems and then combining the components to generate higher-order predictions. Furthermore, in ecology our data of interest are species abundances, resource levels and related variables, rather than energy. To the extent that the hypothesised mathematical structure of $E$ is correct in any given instance, the predictions of these ecological variables will be accurate.

### Deriving species distributions from the cost function

Among the various implementations of DFT, we find that density-potential functional theory (DPFT)[26–31] is uniquely qualified for achieving our objectives. When combined with our energy functional for dispersal (Eq. (2)), DPFT delivers the equilibrium density

$$n_s[V_s - \mu_s](\mathbf{r}) = \frac{1}{\tau_s}[\mu_s - V_s[\mathbf{n}](\mathbf{r})]_+ \qquad (4)$$

of species $s$, where the operator $[\,]_+$ ensures non-negative densities by replacing negative arguments with zero, $\mu_s$ is a species-specific Lagrange multiplier that enforces the density constraint $\int_A (\text{d}\mathbf{r}) \, \mathbf{n}(\mathbf{r}) = \mathbf{N}$ in a focal area $A$ (see Methods), and

$$V_s[\mathbf{n}](\mathbf{r}) = V_s^{\text{env}}(\mathbf{r}) + V_s^{\text{int}}[\mathbf{n}](\mathbf{r}) \qquad (5)$$

is the potential energy as a function of all species (density vector $\mathbf{n}$) at position $\mathbf{r}$, which merges the environment $V_s^{\text{env}}$ (as perceived by species $s$; see Eq. (3)) with the interaction potential $V_s^{\text{int}}[\mathbf{n}]$ (the functional derivative of $E_{\text{int}}[\mathbf{n}]$), coupling $s$ to the other species. The simple relation between density $n_s$ and effective potential $V_s$ in Eq. (4) arises directly from our choice of functional form for the dispersal energy (Eq. (2)). This functional form suffices for the proof-of-principle examples studied here, although other functional forms are also plausible (e.g., Eq. (6) in the Methods). Henceforth, we omit arguments of functions for brevity wherever the command of clarity permits.

The coupled Eqs. (4) and (5) are the essence of DPFT in general and of DFTe in particular. According to Eq. (4), the members of species

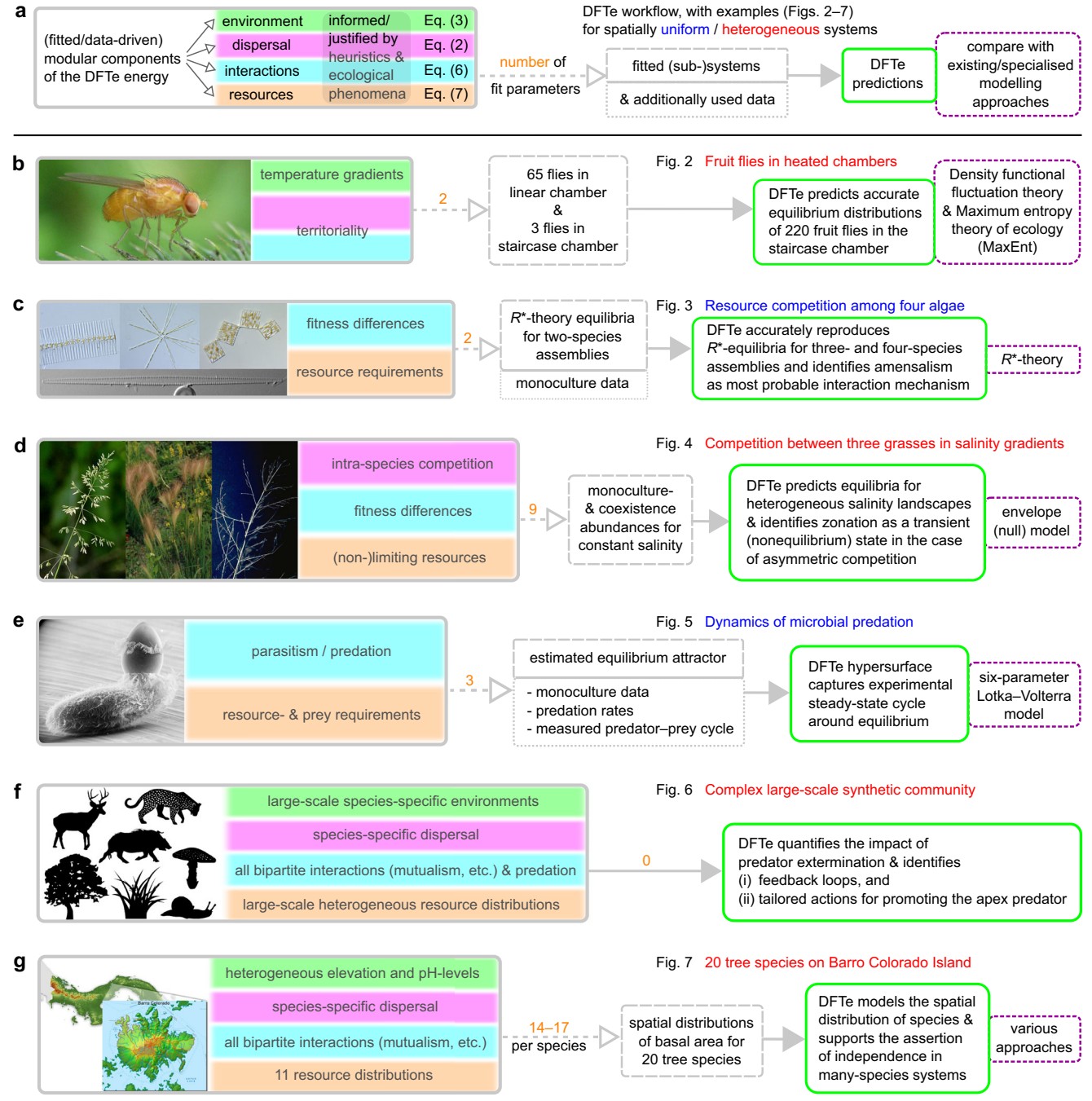

**Fig. 1 | Framework for establishing, validating, and applying DFTe. a** The workflow for extracting DFTe predictions from ecosystems whose properties justify and inform our choices for the components of the DFTe energy functional $E$ (see Methods). Fitting the resulting explicit parameterisations of $E$ to less complex subsystems, we created problem-adjusted tools for addressing the complete systems and their modifications and extensions. We compared DFTe quantitatively with the alternative approaches specified in the violet boxes. **b–g** We highlight our main results for six experimental and synthetic systems (Figs. 2–7), which cover a broad range of taxa, interactions, and environmental settings. The successful modelling of these varied setups gives hope that our DFTe framework can address a

much broader range of ecological systems, and that it is the first stepping stone towards a universal density functional theory for ecology as a whole. In view of the conceptual disparity between DFTe and existing ecological theories, it is expedient to make connections at the level of predictions rather than the basic equations: the DFTe framework's strength is that it brings generality to ecosystem modelling, though it may be outperformed by specialised modelling approaches in specific cases. Details on the parameterisation of each example are given in 'Results' and 'Methods'. We appreciate the photographs of the diatoms, provided by Jason Oyadomari (Fig. 1c, top) and Don Charles (Fig. 1c, bottom); see Supplementary Notes for further details on image sources.

$s$ seek to reside where the cost $V_s$ due to environment and interactions is small: a reduction of $V_s$ increases the density, that is, we interpret small values of $V_s$ as favourable to species $s$. The spatially varying values of $V_s$ themselves are determined through the self-consistent solution of Eqs. (4) and (5). This automatically produces a trade-off

between intra-specific dispersal pressure, which rises with larger density (see Methods), and a confining potential $V_s$ that can be interpreted as an effective environment, where interactions modify the bare environment $V_s^{\mathrm{env}}$. Although this is only one of potentially many ways a computational framework for DFTe could be constructed, it

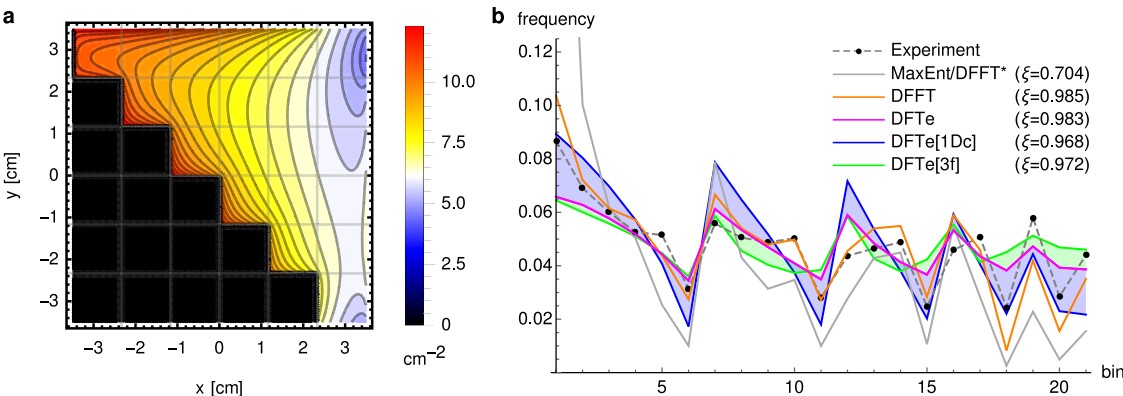

**Fig. 2 | An application to confined fruit flies demonstrates how DFTe can predict the spatial distribution of a population given fixed total abundance. a** The DFTe prediction of the spatial density distribution of 220 interacting flies in a staircase chamber with heat source at $x = 3.5$ cm. The model was parameterised with two datasets[25]: one with 65 flies in a heterogeneously heated elongated chamber [1Dc]; and one with three flies in the staircase chamber [3f]. **b**, The DFTe prediction closely matches the experimental data (each point shows one of the 21 grid cells accessible to flies, in row-major order starting from the top left bin of **a**; overlap of predictions **p** and data **d** is measured with the least-squares correlator $\xi = 2\mathbf{p} \cdot \mathbf{d}/(\mathbf{p}^2 + \mathbf{d}^2)$). The accuracy of DFTe ($\xi \approx 0.98$) is essentially equivalent to DFFT[25]—using the same data for fitting. As expected, DFTe performs more poorly if parameterised based only on the high-density chamber data (DFTe[1Dc]) or only on the low-density staircase data (DFTe[3f]), but any DFTe approach outperforms the naive prediction of maximum entropy theory (MaxEnt/DFFT*).

provides substantial flexibility in terms of the forms of the species-interaction functionals $E_{\text{int}}[\mathbf{n}]$ (see Methods), and we will see that it describes ecological data very well.

### Roadmap for establishing DFTe

In the following, we enlist six biological systems to build and verify the computational DFTe framework. We thereby (i) demonstrate the alignment of DFTe results with both empirical data and existing modelling approaches, (ii) discuss advantages as well as limitations of DFTe relative to existing modelling approaches in ecology, (iii) illuminate how to deploy the building blocks of DFTe in practice, and (iv) show how to extract problem-adjusted DFTe tools from the general DFTe energy for studying modifications and extensions of ecosystems to which our framework has been fit. Figure 1 summarises our agenda by listing the ingredients and main insights of the DFTe predictions for our six systems.

We find that the performance of DFTe is similar to that of existing system-targeted modelling approaches, including density-functional fluctuation theory, the maximum entropy theory of ecology, $R^*$-theory, a climate envelope model, a Lotka–Volterra model, and statistical models. This range of alternative approaches required for modelling our case studies highlights generality as a major benefit of DFTe. We note that specific models constructed within the DFTe framework do not necessarily encode fine-scale causal relations, but may provide effective descriptions, where ecological phenomena cannot always be linked unambiguously to specific DFTe components. The corresponding links declared in Supplementary Table 1 are therefore prototypical rather than rigorous. For example, in our first case study, we may attribute the territoriality of fruit flies either to the local dispersal pressure or to the nonlocal repulsion or some combination of the two (Fig. 1b).

### Fruit flies in heated chambers

To illustrate the fundamental computational procedure of DFTe, i.e., minimisation of the energy functional $E$ for fixed abundances **N**, we used the fruit fly system of Méndez-Valderrama et al.[25]. We found that our DFTe is able to predict distributions of flies in novel situations with similar accuracy to state-of-the-art methods (Fig. 2). Briefly, the DFTe workflow (Fig. 1b) involves fitting a two-parameter energy functional $E$ (Eq. (24) in Methods) to two experimental data sets on fly spatial distributions, one representing responses to high density (crowding) in a heterogeneously heated environment (see Supplementary Fig. 2) and

the other to low density in heterogeneous geometry, and then predicting the response to all factors combined (Fig. 2a). DFTe yields similar accuracies to density-functional fluctuation theory (DFFT) (Fig. 2b), which is developed in ref. [25] as a statistical approach based on the same experimental data but, unlike DFTe, cannot identify mechanisms of the kind implied by Eq. (24). As expected, we found that DFTe performs more poorly if parameterised only on one of the data sets (Fig. 2b), demonstrating that meaningful information on the separate sub-components of the model is extracted from each of the data sets in the full DFTe fit. Although the full DFTe fit is superior to these, any DFTe approach is better than the naive approach of simply scaling up the low-density observations to high density (DFFT* in Fig. 2b), similar to what would be obtained from an approach based on the maximum entropy theory of ecology if solely constrained to the low-density data: Without system-specific information beyond the low-density distribution, a maximum entropy evaluation has to assume independent individuals. This case study illustrates the interplay between dispersal, environment, and interactions in DFTe.

### Resource competition among four algae

Next, we demonstrated how DFTe can incorporate the constraints of resource availability on species' coexistence. In seminal experiments testing predictions of $R^*$-theory, D. Tilman showed that the outcomes of pairwise competition between four algal species can be predicted by fitting a dynamic model to data from single-species experiments[1]. The fitted models can also be used to predict outcomes of multi-species competition, although these have not yet been tested experimentally. We built an alternative tool for predicting the outcomes of multi-species competition by fitting DFTe to predicted pairwise outcomes of algal competition under $R^*$-theory (not Tilman's experimental pairwise outcomes, which in some cases represented transients). We found extremely close correspondence between the predictions of $R^*$-theory and DFTe (Fig. 3; see also Supplementary Table 2). Our method assumed that the algal species' competitive abilities depend on the inverses of the minimum resource requirements ($R^*$-values) and the nutrient requirements per cell (see Methods). Parameterising the DFTe energy components (Eq. (28) in Methods) with species-specific resource requirements and fitness proxies based on Tilman's monoculture data[1], then left just two free parameters to be fit to all pairwise outcomes of resource competition under $R^*$-theory.

This application illustrates the ability of DFTe, despite its very general nature, to scale up from simpler to more-complex systems

with performance similar to more-specialised theories, in this case $R^*$-theory but also a recently proposed community assembly rule that states that the species able to coexist in multi-species competition will be those that can all coexist pairwise[32]. One disadvantage of DFTe relative to $R^*$-theory, however, is that it was parameterised at the two-species level rather than the single-species level, thus avoiding the difficult path of highly parameterised descriptions for multi-species systems at fine scales[33]. We also disregarded the parameters of resource flow rates $\mathbf{F}$ and maximal growth rates in $R^*$-theory: While equilibria for different $\mathbf{F}$ are readily available through $R^*$-theory, they cannot be obtained by the DFTe energy fitted to data for fixed $\mathbf{F}$. This application highlights what in general may be a fruitful path for modelling complex ecological systems: first achieve a detailed understanding in systems with few interacting components (e.g., via $R^*$-theory); then scale up to more-complex systems using DFTe (Fig. 3).

## Competition between three grasses over salinity gradients

We next explored how DFTe can make predictions from sparse information, using an example of three grass species in a saline environment[34]. We fitted the DFTe energy to experimental data on monoculture and mixture densities of the grass species in environments of uniform salinity, and used the fitted model to predict spatially varying densities in a hypothetical environment with heterogeneous salinity (Fig. 4a). We achieved this by making the species' unknown resource requirements parameters that can be fitted to the known data (see Methods). The DFTe predictions were substantially different from a generic envelope model for species distribution modelling (Fig. 4d,e and Supplementary Fig. 7), which simply maps the observed abundances in mixture in different homogeneous environments on to the

heterogeneous environment and, unlike our DFTe model, fails to account for changes in species interactions in the heterogeneous environment. The envelope model's predictions are unlikely to be accurate, because the mixture experiments of ref. [34] already demonstrate that the species do interact in homogeneous environments (mixture abundances are not just rescaled monoculture abundances), and it is likely that these interactions would change in heterogeneous environments. We emphasise, however, that no final verdict on the accuracy of either model, DFTe or the envelope model, can be reached without additional data that test the predictions.

All our examples thus far have focused on equilibria, but time-dependent dynamics are also of great interest to ecologists. We used our case study of competing grasses to illustrate the potential for DFTe to approximate transient dynamics by assuming that the trajectory between the initial and equilibrium states is linear in the total abundance of each species. Using this assumption, we ran a simulation in which two grasses invade the equilibrium monoculture distribution of the third grass (Fig. 4). One phenomenon arising from this simulation is zonation, i.e., the occurrence of sharp boundaries between species distributions despite a continuous environment[35]. We identify zonation as a transient, interaction-induced phenomenon, which in a real system may even persist as an itinerant steady state if environmental disturbances prohibit complete equilibration. This result demonstrates DFTe's ability to produce non-trivial predictions that can be potentially tested with experimental data.

## Dynamics of microbial predation

Our first three examples showed how DFTe can predict equilibria and non-trivial transient dynamics; a fourth example demonstrates that DFTe can also model non-equilibrium steady states, in this case

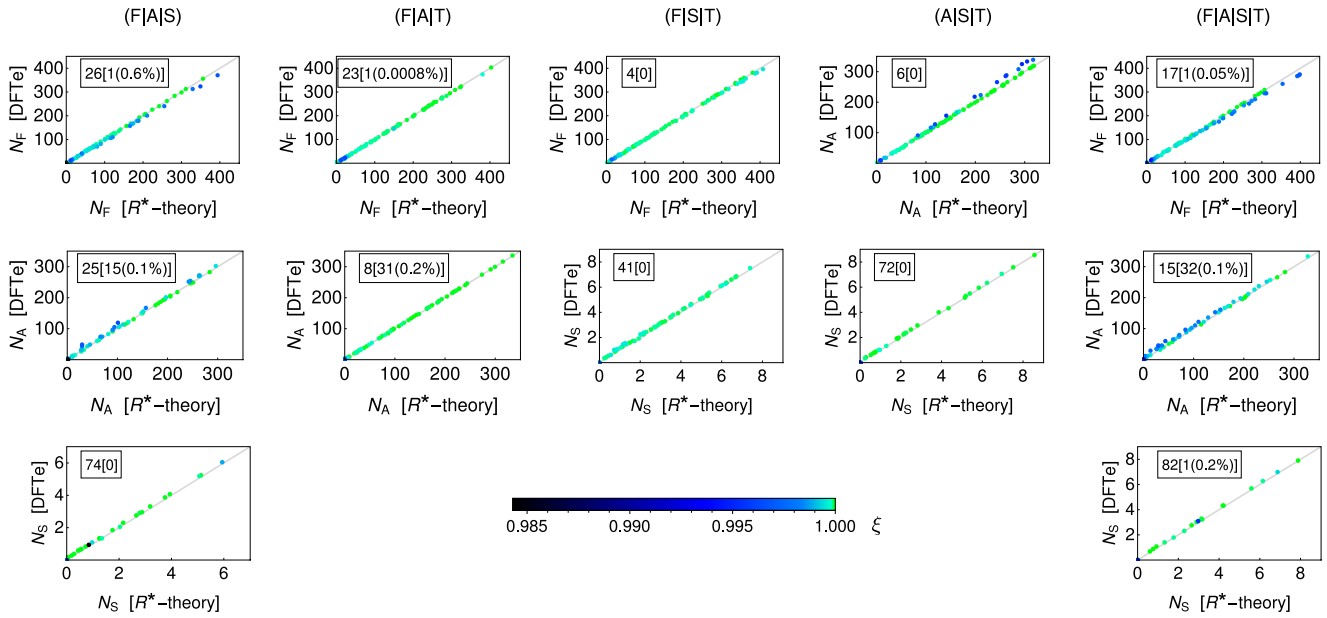

**Fig. 3 | An application to Tilman's algae competition experiments demonstrates DFTe's ability to handle resource constraints.** Without exception, we found that DFTe delivers the $R^*$-equilibria of (i) the four-species assemblage (F|A|S|T) of *Fragilaria crotonensis* (F), *Asterionella formosa* (A), *Synedra filiformis* (S), and *Tabellaria flocculosa* (T), and (ii) the three-species subsets accurately ($\xi \approx 1$; the colour legend applies to all graphs). Each of the five columns of graphs represents 100 randomly drawn resource combinations and shows the resulting DFTe abundances $N_s$ ($s \in \{F, A, S, T\}$; given per micro-litre of suspension) relative to $R^*$-predictions. The DFTe abundances are represented by coloured points, each associated with one resource combination. If both DFTe and $R^*$-theory predict the same abundance of one of the system's species, then the coloured point falls on the

diagonal grey line of the corresponding graph, while its colour reports the overlap $\xi$ of all (three or four) abundances with the $R^*$-predictions for the associated resource combination. We do not show results for T, which is competitively excluded in all 500 cases within both $R^*$-theory and DFTe. The framed box within each graph displays the number of resource cases (out of 100) for which the species was excluded in (i) both $R^*$-theory and DFTe, and (ii) in $R^*$-theory but not in DFTe [in square brackets; the percentage gives the (on-average) abundance of the minority species relative to the majority species, that is, the minority species was almost excluded]. We observed the latter case only if both F and A, which have similar competitive abilities (see Supplementary Notes), were part of the assemblage. Supplementary Fig. 3 shows a summarising histogram in terms of $\xi$.

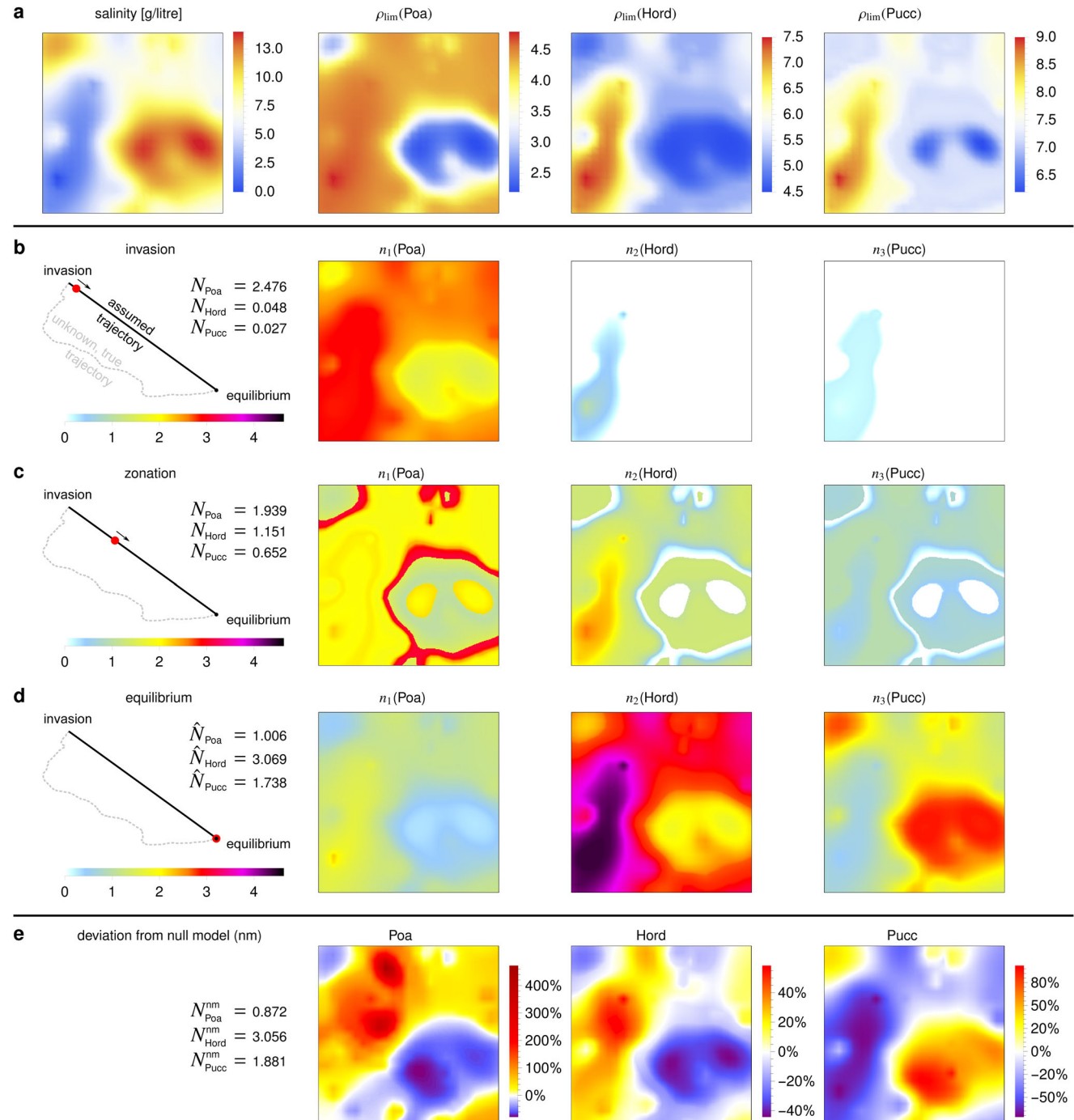

**Fig. 4 | An application to plants in salinity gradients demonstrates how DFTe can extrapolate scarce experimental data into novel more-complex settings. a** We fitted the DFTe functional (Eq. (29) in Methods) with asymmetric repulsive contact interactions between the three grass species *Poa pratensis* (Poa), *Hordeum jubatum* (Hord), and *Puccinellia nuttalliana* (Pucc) to experimental above-ground biomass data. These data are reported in ref. [34] for monoculture and mixture setups at spatially uniform salinity levels. We used the parameterised model to predict spatial distributions of the hypothetical, but data-informed, limiting resources $\rho_{lim}$ for each species in a synthetic landscape with heterogeneous salinity (square area $A = 1$). **b–d** This then allows prediction of the spatial distributions of the densities of

the three species. Panels **b–d** also approximate time-dependent dynamics by showing snapshots of density distributions along the linear trajectory in the space of abundances from an initial state where Poa is in monoculture ($\mathbf{N} = (2.499, 0.0, 0.0)$) (see Supplementary Fig. 4a) to the global equilibrium at $\mathbf{N} = \hat{\mathbf{N}} = (1.006, 3.069, 1.738)$, as indicated by the red dots in the sketches. The model predicts a rich zoo of phases (see Methods and Supplementary Fig. 4), including zonation as a transient state (**c**) on the way to the smooth equilibrated mixture (**d**). The large relative deviations of DFTe densities from those of a generic envelope model ($\mathbf{N}^{nm} = (0.872, 3.056, 1.881)$), which in this case represents a null model, reveal the substantial impact of heterogeneity on the grass distributions (**e**).

occurring in predator–prey dynamics. Our case study here is of *Didinium nasutum* feeding on *Paramecium aurelia*, where experiments have indicated that the long-term steady state can be cycles rather than an equilibrium[36,37] (Fig. 5). We found that the observed

predator–prey cycles are accurately captured by the DFTe energy functional (Eq. (30) in Methods). This shows that the DFTe hypersurface $\mathcal{H}(\mathbf{N})$, which is the space of DFTe equilibria spanned by all permissible $\mathbf{N}$ (see Eq. (14) in Methods) and is specified through system-

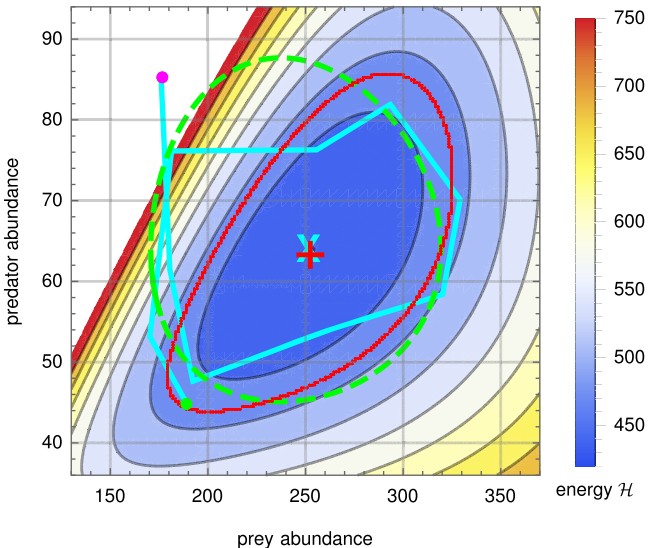

**Fig. 5 | An application to a predator–prey system demonstrates the ability of DFTe to capture non-equilibrium steady-state dynamics.** The DFTe hypersurface $\mathcal{H}(\mathbf{N})$ is a platform for ecosystem dynamics, much like the analogous energy surface of classical physics is for Kepler orbits in a gravitational field. The cyclic DFTe equipotential line (red curve) probes $\mathcal{H}$ away from equilibrium (red plus sign) and captures the empirically measured cycle (cyan curve, starting at the magenta dot and ending at the green dot) centred on the cyan cross, which represents the average measured abundances and anchors the DFTe fit. Strong excitations on $\mathcal{H}$ permit predator extinction along noncyclic trajectories: the high-energy equipotential lines terminate at the abscissa, but not at the ordinate ($\mathcal{H}$ exceeds 750 in the white region and diverges towards the ordinate; see also Supplementary Fig. 8b for an overview plot of $\mathcal{H}$). The DFTe trajectory lacks information on the time-resolved rates of abundance changes and, thus, does not translate unambiguously into a time series that would allow point-wise comparison with predictions from dynamic models—see Supplementary Notes for ways to augment DFTe with explicit Newtonian-type time evolution based on $\mathcal{H}$. However, we compute the DFTe trajectory by fitting only three parameters and find it to be of similar quality to a fitted trajectory from the standard six-parameter Lotka–Volterra model (dashed green; see Supplementary Notes), as judged by $a_{LV} = A_{LV}/A_{exp} - 1 \approx 49\%$ and $a_{DFTe} = A_{DFTe}/A_{exp} - 1 \approx 51\%$, which quantify the excess of the areas $A_{LV}$ and $A_{DFTe}$, enclosed by the corresponding predicted trajectories, over $A_{exp}$, enclosed by the experimental cycle.

specific instances of Eq. (1), has predictive power also away from the equilibrium abundances $\hat{\mathbf{N}}$, although $\hat{\mathbf{N}}$ may a priori follow from a myriad of energy functionals $E[\mathbf{n}, \boldsymbol{\mu}](\mathbf{N})$ that share the same global minimum. The examples of Figs. 4 and 5 are exploratory studies that demonstrate that the DFTe hypersurface can be an appropriate platform for dynamics and that DFTe can potentially be augmented with actual temporal dynamics, thereby opening the door to modelling time-dependent phenomena in ecology with DFTe (see Supplementary Notes).

**A complex large-scale synthetic community**

Next, we explored the full potential of DFTe with the aid of a synthetic community of seven interacting species (Fungus, Deer, Pig, Snail, Tree, Grass, Cat; Fig. 6) subjected to heterogeneous environments and resource distributions intended to represent regional-scale distributions. We selected the parameters of this food web solely for pedagogical reasons, namely to reveal the functioning of all ingredients of the DFTe energy in a controlled way. Using DFTe, we quantified the impact of removing the apex predator and thereby highlighted the capacity of DFTe for predicting the effects of perturbations on complex multi-species communities in complex spatially structured environments[38]. Despite the complexity of this multi-trophic system,

we retain ample intuition of the causal relations between the model input and output by successively building the complete ecosystem. This understanding extends to effects that may appear counter-intuitive when examining only the interactions in Fig. 6a (for example, the small decrease of the Pig abundance upon Cat extermination; see Methods). In a practical setting, this would allow us to promote the apex predator through informed and tailored actions, here for example by reducing the Tree's deforestation stress (see Supplementary Fig. 11). This demonstrates the potential of DFTe for informing management strategies for biodiversity, for which purpose food web models are currently commonly consulted. Our modular and spatially explicit DFTe framework overcomes major challenges of traditional food web analyses, which often lack (i) the crucial incorporation of relevant processes beyond inter-species interactions and (ii) spatial heterogeneity[39], with some exceptions that have been emerging recently (for example, ref. [40]) or that are designed for specific ecosystems (for example, ref. [41]).

**Twenty tree species in a tropical forest**

Finally, we explored the scalability of DFTe towards multi-species systems by modelling the twenty most abundant tree species found in the tropical forest plot on Barro Colorado Island (BCI), Panama[42]. The mechanisms underlying the diversity of tropical forests in general, and the BCI plot in particular, are still not yet fully understood, but have broad implications for ecological theory and policies aimed at maintaining biodiversity[43,44].

Our BCI case study shows how DFTe can deal with an extreme case of missing information, since we use no data beyond the observed data on abundances, soil, and altitude (see Methods). This is not a situation most suited for DFTe models, which are preferably built by determining some parameters from simpler versions of the ecosystem. Here, for example, if the resource requirements were known from monoculture experiments, we could eliminate 220 out of up to 341 fit parameters. High variability arises in the fitted parameters across simulation runs because the large number of energy components of $E$ permits an increase in some components and a decrease in others without significant impact on the density distributions. We can nonetheless draw some general conclusions: the DFTe simulations with and without bipartite interactions yield spatial basal area distributions for each species of similar quality (Fig. 7a–c) despite disparate species-specific fit parameters (Fig. 7d). These results support the hypothesis that inter-species interactions in many-species communities are weak relative to other mechanisms[45]. In the DFTe fit to BCI, the total inter-action energy is relatively small and the extracted interaction kernels (Fig. 7e) are distributed across positive and negative values for all bipartite interactions. Similar conclusions have been reached in previous studies using a specialised multi-scale analysis[44], a maximum entropy approach[46], a stochastic birth–death model[46], and spatial point pattern analysis[47].

## Discussion

Ecology has a long tradition of modelling simple systems with mechanistic equations, for example, of the Lotka–Volterra type. In parallel, large and complex ecological systems have been tackled with statistical approaches. Here we have presented a single framework that facilitates bridging simple and complex models across scales—a goal that has been recognised as both elusive and important for addressing fundamental ecological questions. Inspired by techniques from density functional theory (DFT) in physics, we have created a density functional theory for ecology (DFTe) and demonstrated its alignment with experimental and synthetic data across various ecological systems. No other single theoretical ecological framework has been shown to model such a broad range of systems. We have not yet found a case study of which DFTe does not provide an excellent description, although we encourage further work exploring more case studies.

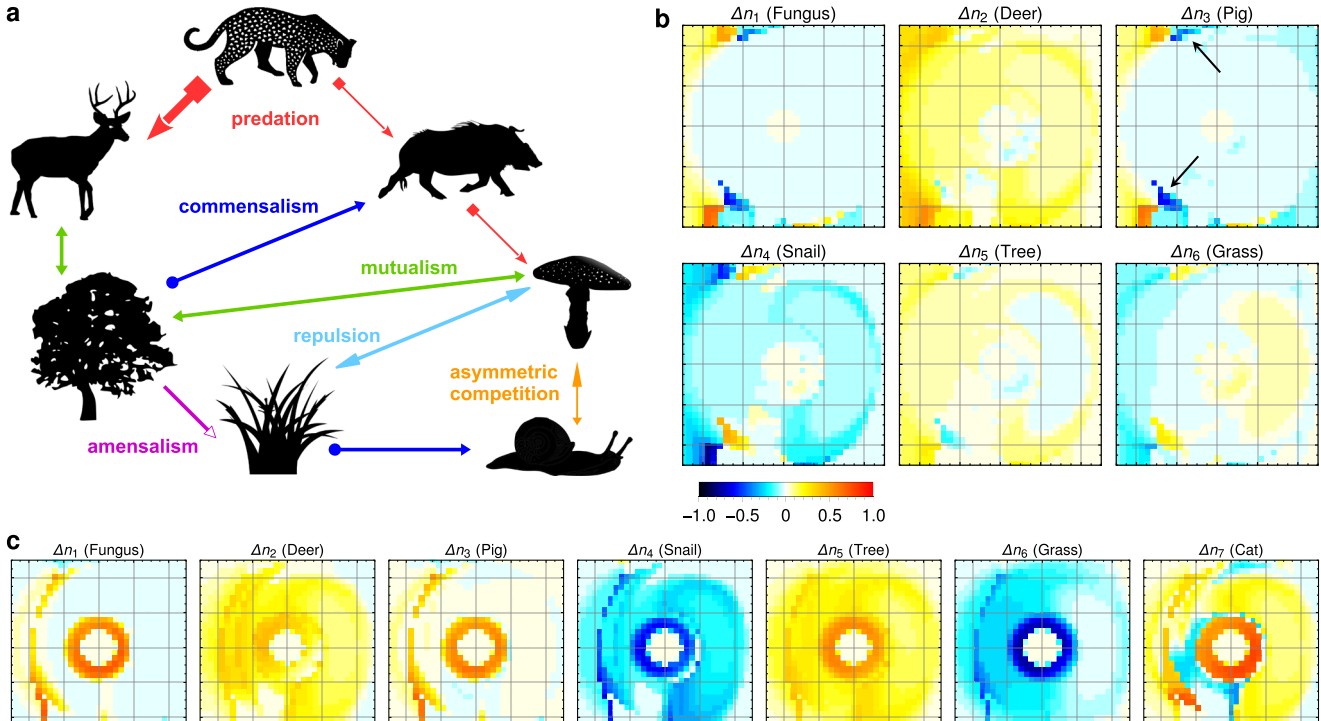

**Fig. 6 | An application to a hypothetical food web demonstrates DFTe's abilities to make predictions about the effects of perturbations on complex ecosystems. a** The cartoon illustrates the food web involving seven hypothetical species on a synthetic landscape with heterogeneous environmental suitability and resource availability (Supplementary Table 4 and Supplementary Fig. 11). **b** The six sub-panels depict the relative changes of species distributions following the extermination of the Cat (see graphical legend; redder colours indicate increasing populations; bluer colours indicate decreasing populations). While density surges of the Cat's prey (the Cat requires both Deer and Pig) are to be expected, the absence of the apex predator has repercussions throughout the entire community —evidently, the interaction-mediated links in the community are strong enough to induce major distortions of the density distributions of all species. This includes effects that may come as a surprise prior to our quantitative simulation, such as

regionally declining Pig populations. Two such enclaves are indicated with arrows. We found the main effect of removing the Cat to be a re-equilibration of the whole ecosystem through feedback loops that promote Fungus, Deer, and Tree, but penalise Grass and Snail. We gained confidence in these DFTe predictions by successively building the complete community from simpler subsystems that permit an intuitive understanding of the relations between the model ingredients and the equilibrated density distributions (see Methods, Supplementary Figs. 12 and 13, and Supplementary Notes). For a real system, we could then use the quantitative knowledge obtained from the perturbation simulation to promote the Cat through informed actions: **c** Cutting the Tree's deforestation stress in half, we increased the Cat population by 36% and created new Cat habitat, especially in the central ring region (see Methods and Supplementary Fig. 14).

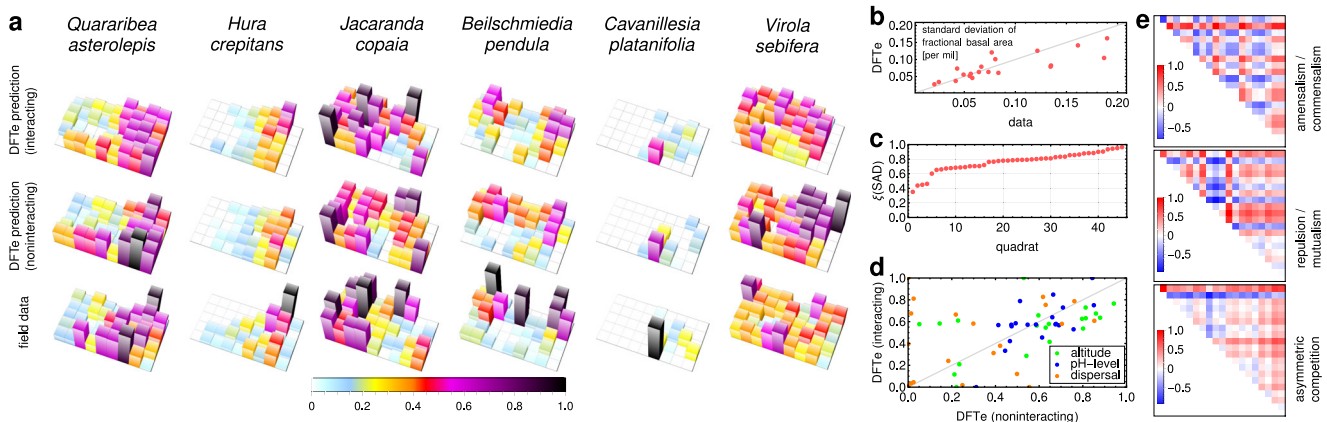

**Fig. 7 | An application to the tropical-forest data from Barro Colorado Island demonstrates the scalability of DFTe towards many-species systems. a** We obtained the DFTe densities $\bar{n}_s\left(\bar{n}_s^0\right)$, which represent the spatially resolved fractional basal area of species $s$ averaged over all available censuses[42], in the top (centre) row of charts by simultaneously fitting the general DFTe energy functional $E$ with (without) inter-species interactions to the reference densities $n_s^{\text{ref}}$ in the bottom charts; see Methods. Both $\bar{n}_s$ and $\bar{n}_s^0$ are an adequate fit ($\xi \approx 0.79$) to $n_s^{\text{ref}}$ for all twenty species; see also Supplementary Fig. 15 and Supplementary Table 5. In the

three charts for each of the species shown ($s = 1, 4, 8, 12, 16, 20$) we report the density values relative to $\max_{\mathbf{r}}\{\bar{n}_s(\mathbf{r}), \bar{n}_s^0(\mathbf{r}), n_s^{\text{ref}}(\mathbf{r})\}$. Complementing measures of the quality of $\bar{\mathbf{n}}$ are the species-resolved variability in density (**b**) and the (sorted) least-squares overlaps $\xi$ of abundances $\mathbf{N}(Q)$ with $\mathbf{N}^{\text{ref}}(Q)$, viz. the overlaps of local species-abundance distributions (SAD) for each of the 45 quadrats $Q$ (**c**). **d** The fit parameters associated with altitude, pH-level, and dispersal, with and without bipartite interactions. **e**, The fit results for the bipartite interaction kernels (see Methods); each sub-panel is normalised to its respective maximum value.

The key advantage of DFTe comes from its modularity: the energy functional comprises multiple components, each of which has an intuitive ecological interpretation and each of whose parameters can be inferred from simpler subsystems. Another advantage is that the complexity of the DFTe energy functional, which mirrors that of the target ecosystem, is of little technical concern for retrieving the density distributions and the trade-offs between the involved processes. This resolves the ubiquitous problem of finding the proper trade-offs among given mechanisms and constraints for many species that interact within a complex environment. The ecological constituents and mechanisms are independently parameterised components of, and coupled through, the DFTe functional, whose composite structure permits hassle-free revision and introduction of causal mechanisms that can correct disparities between predictions and data. For example, higher-order interactions[48,49] can simply be added to Eq. (9) in the Methods. Densities and abundances may refer to individuals or aggregated observables like biomass or land cover. Conversely, we may fork one species into many, for example, to account for heterogeneity among individuals[9,50].

In the degree of mechanism incorporated, DFTe is intermediate among past approaches to theoretical ecology. It is more mechanistic than statistical or correlative approaches such as MaxEnt. It explicitly encodes processes like predation, competition, and consumption, and it uses individual-level parameters such as individual nutrient requirements and can produce predictions for systems at fine scales. (By analogy, in physics Newton's law of gravity, a prototypical example of a mechanistic model, aggregations of elementary particles, that is, density distributions $n(\mathbf{r})$, yield the external/environmental energy $\int (d\mathbf{r})\, V^{gravity}(\mathbf{r})\, n(\mathbf{r})$, the exact analogue of Eq. (3).) But DFTe is more phenomenological than, for example, $R^*$-theory. Indeed, in our application to algal competition, DFTe takes outputs from $R^*$-theory as input for addressing more complex situations (Fig. 3). In common with all other existing ecological theories, DFTe is not fully mechanistic in the sense of a yet-to-be-discovered fundamental (first-principles) framework at the level of individuals from which DFTe might emerge, since its components are not derived from individual-level interactions. This intermediate representation of mechanism in DFTe is a strength of our approach, because while existing mechanistic approaches become increasingly intractable with rising system complexity[3] and while statistical approaches such as MaxEnt—by design—make no attempt at interpreting the data, DFTe can be scaled to complex systems without sacrificing a degree of mechanistic understanding.

One important consideration when applying DFTe is how to deal with data scarcity. Here again DFTe's performance is intermediate between past mechanistic and statistical approaches—the latter of which perform particularly poorly in data-poor situations. Important practical challenges towards a universally useful DFTe in this regard are (i) the selection of the most important components of the energy functional, which is expedient for including as few fitted parameters as possible, and (ii) the required basic intuition on how to combine the ecologically relevant parameters into an energy functional expression. With our range of examples (summarised in Fig. 1), we have highlighted that rudimentary insight into ecosystem functioning can be sufficient for generating results on par with established approaches. Additional illustrations of the intuition behind constructing energy functionals for ecological phenomena are provided in the Supplementary Notes. We judge the main technical challenge in future applications of DFTe to be the parameterisation of models at landscape scales relevant to many ecological applications. Such applications will require not only more data and careful choice of parsimonious functional forms for the energy components, but also high-performance computing.

We have demonstrated the potential for DFTe to provide a unified modelling framework for ecology via applications to a variety of systems, from predatory microbes in test tubes to interacting grasses in saline landscapes to a synthetic food web. Our approach is inspired by DFT in physics, with the key difference that, whereas the DFT energy functional is derived from first principles, the DFTe energy functional must be established via ecological intuition and parsimony. Ultimately, as with any ecological model, the arbiter of the value of DFTe must be determined by its ability to fit and predict ecological data, and on this count our applications have shown DFTe to be comparable to existing specialised approaches. While we are optimistic that DFTe can become a universal tool for an intuitive approach to ecology as a whole, including dynamics on all relevant time scales (see Supplementary Notes), only in-depth future analyses can be the ultimate judge on that matter. History provides vivid examples of physics-inspired models that have led to powerful frameworks in other fields, such as the gravity model of human mobility, whose initial heuristic foundations have been refined over decades towards a parameter-free model[51].

On a general note, we hope that our foundational work at the cross-roads of physics and ecology inspires researchers and practitioners across disciplines towards developing DFT-based tools that tackle their big questions, such as pandemics, traffic control, or the dynamics of financial assets. These are serious enterprises, and a concerted effort will be required. We hope to start an interdisciplinary dialogue that transfers one of the most powerful methods of physics to unexpected realms.

## Methods
### Modular components of the DFTe energy functional
The central ingredient of DFTe is an energy functional $E$, assembled according to Eq. (1). The methodology of DFTe can be understood by inspecting the dispersal and environmental energies in Eqs. (2) and (3) without interactions. In our first case study, illustrated in Fig. 2 and Supplementary Fig. 2, we demonstrate that equation (3), in conjunction with Eq. (2), can realistically describe the influence of the environment on species' distributions. Mechanisms that alter the trade-off between dispersal and environment can be introduced as part of $E_{int}$. For instance, back reactions on the environment could be modelled with a bifunctional $E_{br}[\mathbf{V}^{env}, \mathbf{n}]$ that yields the equilibrated modified environment $V_s^{env} + \delta E_{br}[\mathbf{V}^{env}, \mathbf{n}]/\delta n_s(\mathbf{r})$, cf. Eq. (5).

In the following we make explicit the interaction and resource energies that enter Eq. (1) and are used in our case studies of Figs. 2–7. We let $E_{int}[\mathbf{n}]$ include all possible bipartite interactions

$$E_\gamma[\mathbf{n}] = \sum_{\substack{s,s'=1 \\ s' \neq s}}^{S} \int_A (d\mathbf{r})(d\mathbf{r}')\, n_s(\mathbf{r})^{\alpha_s}\, \gamma_{ss'}(\mathbf{r}, \mathbf{r}')\, n_{s'}(\mathbf{r}')^{\beta_{s'}}, \qquad (6)$$

which include amensalism, commensalism, mutualism, and so forth. Here, $\alpha_s, \beta_{s'} \geq 0$, and the interaction kernels $\gamma_{ss'}$ are assembled from fitness proxies of species $s$ and $s'$ (Supplementary Table 1). Higher-order interactions can be introduced, for example, through (i) terms like $n_s\, \gamma_{ss'}\, n_{s'}\, \gamma'_{s's''}\, n_{s''}$ that build on pairwise interactions or (ii) genuinely multipartite expressions like $\gamma_{ss's''} n_s\, n_{s'}\, n_{s''}$. Multi-partite interactions based on bipartite interactions do not seem to be an uncommon scenario[48]. However, there may be systems where nonzero coefficients $\gamma_{ss''}$ couple all species. This poses a challenge for mechanistic theories in general. Then, 'simpler subsystems' that have to be included in the DFTe workflow of Fig. 1a can only refer to situations where other energy components are absent, such as resource terms or complex environments. For example, the coefficients $\gamma_{ss''}$ could be extracted in an experiment with a controlled simple environment and then used to model the interacting species in a real-world setting. For $(\alpha_s, \beta_{s'}) = (1,1)$ we identify the contact interaction in physics as $\gamma_{ss'} \propto \delta(\mathbf{r} - \mathbf{r}')$ with the two-dimensional delta function $\delta(\ )$, while the Coulomb interaction amounts to setting $\gamma_{ss'} \propto 1/|\mathbf{r} - \mathbf{r}'|$. The mechanistic effect of these interaction kernels on the density distributions is the same in ecology as it is in physics—a mathematical insight that inspired us to build

ecological analogues to the phenomenology of quantum gases, which feature functionals of the kind in Eq. (6). Note that we do not introduce any quantum effects into ecology despite the fact that the mathematical structure of DFTe is borrowed in part from quantum physics. While the contact interaction is a suitable candidate for plants and especially microbes[52], we expect long-range interactions (for example, repulsion of Coulomb type) to be more appropriate for species with long-range sensors, such as eyes. Both types of interactions feature in describing the ecosystems addressed in this work.

In a natural setting the equilibrium abundances are ultimately constrained by the accessible resources. It is within these limits of resource availability that environment as well as intra- and inter-specific interactions can shape the density distributions. An energy term for penalising over- and underconsumption of resources is thus of central importance. Each species consumes resources from some of the $K$ provided resources, indexed by $k$. A subset of species consumes the locally available resource density $\rho_k(\mathbf{r})$ according to the resource requirements $\nu_{ks}$, which represent the absolute amount of resource $k$ consumed by one individual (or aggregated constituent) of species $s$. The simple quadratic functional

$$E_{\mathrm{Res}}[\mathbf{n}] = \int_A (\mathrm{d}\mathbf{r}) \sum_{k=1}^{K} \mathcal{L}_k(\mathbf{n},\rho_k) \equiv \zeta \int_A (\mathrm{d}\mathbf{r}) \sum_{k=1}^{K} w_k(\mathbf{r}) \left[ \sum_{s=1}^{S} \nu_{ks} n_s(\mathbf{r}) - \rho_k(\mathbf{r}) \right]^2 \tag{7}$$

proves appropriate. Here, $\nu_{ks} n_s$ is the portion of resource density $\rho_k$ that is consumed by species $s$. That is, $\nu_{ks} > 0$ indicates that $s$ requires resource $k$. If Eq. (7) is the total energy functional, then a single-species system with a single resource equilibrates with density $n_1(\mathbf{r}) = \rho_1(\mathbf{r})/\nu_{11}$ at every position $\mathbf{r}$, and additional DFTe energy components would modify this equilibrium. Predator–prey relationships are introduced by making species $k$ a resource $\rho_k = \,]n_k[$, where $]n[$ declares $n$ a constant w.r.t. the functional differentiation of $E$, that is, the predator tends to align with the prey, not the prey with the predator. In view of the energy minimisation, the quadratic term in Eq. (7) entails that regions of low resource density $\rho_k$ are less important than regions of high $\rho_k$. The different resources $k$ have the same ability to limit the abundances, such that the limiting resource $k = l$ at $\mathbf{r}$ has to come with the largest of weights $w_l(\mathbf{r})$, irrespective of the absolute amounts of resources at $\mathbf{r}$. For example, the weights $w_k$ have to ensure that an essential but scarce mineral has (a priori) the same ability to limit the abundances as a resource like water, which might be abundant in absolute terms. To that end, we specify the weights

$$w_k(\mathbf{r}) = \frac{1}{\bar{\rho}_k^2} \sum_{s} \eta(\nu_{ks}) \exp\left[\sigma\left(\frac{\lambda_{ks}}{\lambda_{ls}} - 1\right)\right], \tag{8}$$

which are inspired by the smooth minimum function, where $\sigma < 0$, $\lambda_{ks}(\mathbf{r}) = \rho_k(\mathbf{r})/\nu_{ks}$, and the carrying capacity of the limiting resource is $\lambda_{ls} = \min_k \lambda_{ks}$. The step function $\eta()$ in Eq. (8) ensures that only resources $k$ that are actually consumed by species $s$ contribute to $w_k$. We rescale $w_k$ using the average $\bar{\rho}_k = \frac{1}{A} \int_A (\mathrm{d}\mathbf{r}) \rho_k(\mathbf{r})$, which puts all resources on equal footing in their ability to limit abundances and renders $E_{\mathrm{Res}}$ invariant under change of resource units. The ratio of carrying capacities in Eq. (8) implements the relative importance of all resources at $\mathbf{r}$, with resources $k \neq l$ suppressed exponentially according to their deviation from their ability to limit the abundances. For example, with $\sigma = -4$ and $\lambda_{ks} = 2\lambda_{ls}$, resource $k \neq l$ is largely irrelevant for species $s$ as it weighs in at less than 2% compared to the limiting resource $l$. There can be multiple limiting resources for a species $s$, most conceivably, multiple resources that vanish locally. In the limit of vanishing resource density ($\rho_l, \lambda_{ls} \to 0$), the exponential in Eq. (8) reduces to the Kronecker-delta $\delta_{kl}$, thereby rendering all resources with $\lambda_{ks} > \lambda_{ls}$ irrelevant at $\mathbf{r}$. Using $E_{\mathrm{Res}}$, we show that an analytically solvable minimal example of two amensalistically interacting species

already exhibits a plethora of resource-dependent equilibrium states (see Supplementary Notes and Supplementary Fig. 1).

We specify the DFTe energy functional in Eq. (1) by summing Eqs. (2), (3), (6), and (7) and by (optionally) constraining the abundances to $\mathbf{N}$ via Lagrange multipliers $\boldsymbol{\mu}$:

$$E[\mathbf{n},\boldsymbol{\mu}](\mathbf{N}) \equiv E[\mathbf{n}] + E_{\boldsymbol{\mu}}[\mathbf{n}](\mathbf{N})$$

$$\equiv E_{\mathrm{dis}}[\mathbf{n}] + E_{\mathrm{env}}[\mathbf{n}] + E_\gamma[\mathbf{n}] + E_{\mathrm{Res}}[\mathbf{n}] + \sum_{s=1}^{S} \mu_s \left( N_s - \int_A (\mathrm{d}\mathbf{r})\, n_s \right). \tag{9}$$

Uniform situations are characterised by spatially constant ingredients $n_s = N_s/A$, $\rho_k = R_k/A$, coefficients $\tau_s$, etc. for the DFTe energy, such that Eq. (9) reduces to a function $E(\mathbf{N})$ with building blocks

$$E_{\mathrm{dis}} \longrightarrow \frac{1}{2A} \sum_{s=1}^{S} \tau_s N_s^2, \tag{10}$$

$$E_{\mathrm{env}} \longrightarrow \sum_{s=1}^{S} V_s^{\mathrm{env}} N_s, \tag{11}$$

$$E_\gamma \longrightarrow \sum_{\substack{s,s'=1 \\ s' \neq s}}^{S} \frac{N_s^{\alpha_s} \gamma_{ss'} N_{s'}^{\beta_{s'}}}{A^{\alpha_s + \beta_{s'} - 1}}, \tag{12}$$

$$E_{\mathrm{Res}} \longrightarrow A \sum_{k=1}^{K} \mathcal{L}_k(\mathbf{N}/A, R_k/A). \tag{13}$$

## Ecosystem equilibria from the DFTe energy functional

The general form of Eq. (9) gives rise to two types of minimisers (viz., equilibria): First, we term

$$\mathcal{H}(\mathbf{N}) \equiv E[\tilde{\mathbf{n}}] \equiv \min_{\mathbf{n}} \left\{ E[\mathbf{n}] \, \Big| \, \int_A (\mathrm{d}\mathbf{r})\, \mathbf{n}(\mathbf{r}) = \mathbf{N} \,(\mathrm{fixed}) \right\} \tag{14}$$

the 'DFTe hypersurface', with $\tilde{\mathbf{n}}$ the energy-minimising spatial density profiles for given (fixed) $\mathbf{N}$. Second, the ecosystem equilibrium is attained at the equilibrium abundances $\hat{\mathbf{N}} = \int_A (\mathrm{d}\mathbf{r}) \hat{\mathbf{n}}(\mathbf{r})$, which yield the global energy minimum

$$\mathcal{H}(\hat{\mathbf{N}}) = \min_{\mathbf{N}} \mathcal{H}(\mathbf{N}), \tag{15}$$

where the minimisation samples all admissible abundances, that is, $\mathbf{N} \in \left(\mathbb{R}_0^+\right)^{\times S}$ if no further constraints are imposed.

The direct minimisation of $E[\mathbf{n}]$ is most practical for uniform systems, which only require us to minimise $E(\mathbf{N})$ over an $S$-dimensional space of abundances. For the general nonuniform case, we adopt a two-step strategy that reflects Eqs. (14) and (15). First, we obtain the equilibrated density distributions on $\mathcal{H}$ for fixed $\mathbf{N}$ from the computational DPFT framework[26–31]. Second, a conjugate gradient descent searches $\mathcal{H}(\mathbf{N})$ for the global minimiser $\hat{\mathbf{N}}$. Technically, we perform the computationally more efficient descent in $\boldsymbol{\mu}$-space. Local minima are frequently encountered, and we identify the best candidate for the global minimum from many individual runs that are initialised with random $\boldsymbol{\mu}$. Note that system realisations with energies close to the global minimum, especially local minima, are likely observable in reality, assuming that the system can equilibrate at all. There is always an equilibrium if the energy functional is bounded from below, together with the fact that the support (abundances/densities) of the energy functional is finite in any practical application. If some DFTe energy components are chosen (too) negative, the system can be unstable, in which case the energy functional has no minimum and is inappropriate for modelling the equilibrium in question. This means that another

energy functional has to be considered, or, in the worst case, that DFTe is incapable of simulating this system. We also caution that no numerical optimisation algorithms for non-convex black-box functions can guarantee to find the global minimum, not even approximately. Without analytically available characteristics of the global minimum, all one may hope for are candidates of the minimiser, and those may not even be local minima−there is no way to be certain that an optimum proposed by a numerical optimisation algorithm is stable.

## Density-potential functional theory (DPFT) in Thomas−Fermi (TF) approximation

Defining

$$V_s(\mathbf{r}) = \mu_s - \frac{\delta E_{\mathrm{dis}}[\mathbf{n}]}{\delta n_s(\mathbf{r})} \tag{16}$$

for all $s$, we obtain the reversible Legendre transform

$$E_{\mathrm{dis}}^{\mathrm{L}}[\mathbf{V} - \boldsymbol{\mu}] = E_{\mathrm{dis}}[\mathbf{n}] + \sum_{s=1}^{S} \int_A (\mathrm{d}\mathbf{r}) (V_s - \mu_s) n_s \tag{17}$$

of the dispersal energy and thereby supplement the total energy with the additional variables $\mathbf{V}$:

$$E[\mathbf{V}, \mathbf{n}, \boldsymbol{\mu}](\mathbf{N}) = E_{\mathrm{dis}}^{\mathrm{L}}[\mathbf{V} - \boldsymbol{\mu}] - \int_A (\mathrm{d}\mathbf{r}) \, \mathbf{n} \cdot (\mathbf{V} - \mathbf{V}^{\mathrm{env}}) + E_{\mathrm{int}}[\mathbf{n}] + \boldsymbol{\mu} \cdot \mathbf{N}. \tag{18}$$

This density-potential functional is equivalent to (but more flexible than) the density-only functional $E[\mathbf{n}, \boldsymbol{\mu}](\mathbf{N})$. The minimisers of $E[\mathbf{n}]$ are thus among the stationary points of Eq. (18) and are obtained by solving

$$n_s[V_s - \mu_s](\mathbf{r}) = \frac{\delta E_{\mathrm{dis}}^{\mathrm{L}}[V_s - \mu_s]}{\delta V_s(\mathbf{r})} \tag{19}$$

and

$$V_s[\mathbf{n}](\mathbf{r}) = V_s^{\mathrm{env}}(\mathbf{r}) + \frac{\delta E_{\mathrm{int}}[\mathbf{n}]}{\delta n_s(\mathbf{r})} \tag{20}$$

self-consistently for all $n_s$ while enforcing $\int_A (\mathrm{d}\mathbf{r}) \, n_s(\mathbf{r}) = N_s$. Specifically, starting from $\mathbf{V}^{(0)} = \mathbf{V}^{\mathrm{env}}$, such that $n_s^{(0)} = n_s[V_s^{(0)} - \mu_s^{(0)}]$, we iterate

$$n_s^{(i)} \xrightarrow{\text{equation (20)}} V_s^{(i+1)} = V_s[\mathbf{n}^{(i)}] \xrightarrow{\text{equation (19)}} n_s^{(i+1)} = (1 - \theta_s) n_s^{(i)} + \theta_s \, n_s\big[V_s^{(i+1)} - \mu_s^{(i+1)}\big] \tag{21}$$

until all $n_s$ are converged sufficiently. This self-consistent loop establishes a trade-off between dispersal energy and effective environment $\mathbf{V}$ by forcing an initial out-of-equilibrium density distribution to equilibrate at fixed $\mathbf{N}$. We adjust $\mu_s^{(i)}$ in each iteration $i$ such that $n_s^{(i)}$ integrates to $N_s$. Small enough density admixtures, with $0 < \theta_s < 1$, are required for convergence. Each point of $\mathcal{H}(\mathbf{N})$ in Eq. (14) represents the thus-equilibrated system for given $\mathbf{N}$. The dispersal energy $E_{\mathrm{dis}}[n_s]$ in Eq. (2), where an increase of the dispersal pressure constants $\tau_s$ tends to dilute $n_s$ (see also Supplementary Fig. 2), is only one component of the total energy and has to be balanced against, for example, the environmental energy. This trade-off produces the aggregation of, for example the fruit flies in Fig. 2 and the grasses in Fig. 4. In the latter case the confining role is played by the interaction potential, not the environmental potential, which are combined anyway through Eq. (5). If $E_{\mathrm{dis}}$ with positive $\tau_s$ were the only energy component in an unconstraining, i.e., flat and infinite environment, then $\min_{n_s} E_{\mathrm{dis}}[n_s] = 0$ for a density that is maximally dispersed/diluted with $n_s(\mathbf{r}) \to 0$ everywhere.

Assuming a given interaction energy, we require an explicit expression for the right-hand side of Eq. (19) to realise the self-consistent loop of Eq. (21). For two-dimensional fermion gases in TF approximation[28], we have

$$E_{\mathrm{dis}}^{\mathrm{L}}[\mathbf{V} - \boldsymbol{\mu}] = -\frac{1}{2} \sum_{s=1}^{S} \int (\mathrm{d}\mathbf{r}) \frac{1}{\tau_s} [\mu_s - V_s]_+^2, \tag{22}$$

which delivers (i) the density in Eq. (4) to be used for the right-hand side of Eq. (19) and (ii) Eq. (2) via Eq. (17) upon inverting the functional relationship $n[V]$ of Eq. (4). For stabilising the numerics if necessary or if, for example, unambiguous derivatives of the density are sought, we replace Eq. (4) by its smooth version

$$n_s[V_s - \mu_s](\mathbf{r}, T) = \frac{1}{\tau_s} \big[\mu_s - V_s\big]_T, \tag{23}$$

where $[x]_T = T \log\big[1 + \exp(x/T)\big]$ is a smooth version of $[x]_+$. If $\tau_s = 0$, we can add a dispersal energy with positive $\tau_s$ to $E$ in order to execute the self-consistent loop with Eqs. (4) or (23), and compensate by subtracting the same dispersal energy from the interaction energy, which enters Eq. (20). The notation $n_s[V_s - \mu_s](\mathbf{r})$ declares that $n_s$ is a functional (function of functions) of $V_s - \mu_s$, and that $n_s$ is also a function of $\mathbf{r}$. In the specific case of Eq. (4), the functional dependence of $n_s$ reduces to a trivial dependence on $V_s - \mu_s$, which renders $n_s$ dependent on $\mathbf{r}$ alone−a consequence of Eq. (22) as the source of Eq. (4). Alternatives to Eq. (22) with nonlocal integrands will make $n_s$ a nontrivial functional of $V_s - \mu_s$, for example, $n_s[V_s - \mu_s](\mathbf{r}) = \frac{1}{\tau_s} \int (\mathrm{d}\mathbf{r}') \frac{[\mu_s - V_s(\mathbf{r})]_+}{|\mathbf{r} - \mathbf{r}'|}$, which reduces to Eq. (4) if $1/|\mathbf{r} - \mathbf{r}'|$ is replaced by $\delta(\mathbf{r} - \mathbf{r}')$.

## Explicit parameterisations of the DFTe energy functional

We model ecological phenomena with the help of the potentials introduced in Eqs. (5) and (20) as a consequence of the energy functional. They are mathematical constructs that can be measured and validated only through a mathematical relation with observable quantities like densities and abundances. At best, we can claim that these mathematical constructs describe our observations appropriately. The situation is no different in physics, see Supplementary Notes. We also note that all DFTe parameters of the specific data-driven models pertinent to Figs. 2–5 and 7 are either fit parameters or are inferred from the data.

**Parameterisations for the fruit flies in heated chambers.** In view of Eq. (4), an educated guess informed by the measured reference density $n_{\mathrm{ref}}^{\mathrm{1Dc}}$ (see Supplementary Fig. 2) is a quadratic environment induced by the heat source at $x_0$. We also include a repulsive long-range interaction, since (i) the data in Fig. 2g of ref. [25] suggests that the interaction is quadratic in the density and (ii) fruit flies have to sense their peers remotely, for example, in defending territory[53]. This leaves us with two parameters, $\varepsilon$ and $\gamma$, for fitting the minimising density of

$$E[n, \mu](N) = \frac{1}{2} \int_A (\mathrm{d}\mathbf{r}) \, n(\mathbf{r})^2 + \varepsilon \int_A (\mathrm{d}\mathbf{r})(x + x_0)^2 \, n(\mathbf{r}) + \gamma \int_A (\mathrm{d}\mathbf{r})(\mathrm{d}\mathbf{r}') \frac{n(\mathbf{r}) n(\mathbf{r}')}{|\mathbf{r} - \mathbf{r}'|}$$
$$+ \mu \left( N - \int_A (\mathrm{d}\mathbf{r}) \, n(\mathbf{r}) \right) \tag{24}$$

to $n_{\mathrm{ref}}^{\mathrm{1Dc}}$. In the Supplementary Notes we offer a more detailed account of the reasoning behind Eq. (24). In ref. [25] the interaction parameters from the quasi-one-dimensional (1D) chamber are combined with the environmental information from a low-density experiment (three flies) in a 'staircase chamber' to predict a high-density distribution of 220 flies (labelled 'DFFT' in Fig. 2b). This is a reasonable procedure, since the low-density three-flies experiment represents a situation with very

small total repulsion and is therefore well suited for extracting the environmental influence on the density distribution. We thus follow the same strategy. Keeping $\gamma = 9\,\text{cm}$ from the density fit to the quasi-1D chamber setup, we get $\varepsilon = 14.5\,\text{cm}^{-2}$ for three flies in the staircase chamber and use both parameters for predicting the density distribution of 220 flies (labelled 'DFTe' in Fig. 2b). Since $\gamma$ is (i) one of the system's characteristic length scales, (ii) an estimator of the spatial reach of the long-range interaction, and (iii) close to the system size, we deem the fruit flies setup a small-scale system—in contrast to the large-scale contact-interacting systems of Figs. 4 and 6.

**Parameterisations for the resource competition among four algae.** We face a uniform environment with $S$ species and two resources. An amensalistic interaction is an appropriate candidate for putting a subset of species at a disadvantage relative to their heterospecifics (see Supplementary Table 1). We rule out parasitism, understood in the sense of an asymmetric interaction, for which the DFTe equilibria do not produce the correct survivors in all reference resource cases $\mathcal{R}_{1-7}$ (see Supplementary Table 2). In contrast to the fruit-flies study above, the abundances are not fixed but are rather to be determined as minimisers of the $\mathbf{N}$-dependent energy function $E(\mathbf{N}) = E_{\text{Res}}(\mathbf{N}) + E_\gamma(\mathbf{N})$, assembled from Eqs. (10)–(13). The spatially uniform interaction kernel

$$\gamma_{ss'} = \gamma \left\{ \left[ \left( \frac{f_s}{f_{s'}} \right)^\kappa - 1 \right]_+ + \left[ \left( \frac{g_s}{g_{s'}} \right)^{1/\kappa} - 1 \right]_+ \right\} \quad (25)$$

for $E_\gamma(\mathbf{N})$, see Eq. (12), is assembled from the fitness proxies

$$f_s = \sum_{k=1}^{2} \frac{w_k}{R_{sk}^*} \quad (26)$$

and

$$g_s = \sum_{k=1}^{2} \frac{w_k}{\nu_{ks}}, \quad (27)$$

which we assume to influence the species' ability to consume the provided resources in the presence of heterospecifics. In the Supplementary Notes we provide a complementary step-by-step account of the intuition that leads us to Eqs. (25)–(28). The $s$-specific traits $R_{sk}^*$ (the densities of resource $k$, below which species $s$ cannot survive in monoculture) and $\nu_{ks}$ (see Eq. (7)) follow from monoculture experiments (see Supplementary Notes and Table 1 of ref. [1]). The exponent $\kappa$ in Eq. (25) introduces a hierarchy between the fitness proxies of Eqs. (26) and (27) and serves as a second parameter to be used in the fitting of our reference data, which are all 42 'reference' abundances $\mathbf{N}_\mathcal{R}$ of two-species $R^*$-equilibria that follow from $\mathcal{R}_{1-7}$. The parameter $\gamma$ enables the trade-off between competition and resource energy. In the spirit of $R^*$-theory, we choose $\sigma \to -\infty$ in Eq. (8), such that only the limiting resource for each species enters the weights $w_k$ for $E_{\text{Res}}$. Finally, the minimisers of

$$E(\mathbf{N}) = \sum_{k=1}^{2} w_k \left[ \sum_{s=1}^{S} \nu_{ks} N_s - \rho_k \right]^2 + \sum_{\substack{s,s'=1 \\ s' \neq s}}^{S} \gamma_{ss'} N_{s'} \quad (28)$$

yield the best fit to $\mathbf{N}_\mathcal{R}$ for $\gamma = 8 \times 10^{-8}$ and $\kappa = 8$, used for modelling the three- and four-species communities in Fig. 3.

It would be surprising if Eq. (25) were the only possible choice for the interaction kernel, and it is quite likely that many different energy functionals that encode different mechanisms and input parameters are equally suited for reproducing the data targeted in this example. Only future DFTe studies that take into account additional data can reduce the space of possible functionals. Taking the cue from physics, we may

hope that a universal functional will prevail and thereby provide deeper insights into the fundamental mechanisms and relations underlying ecological systems in general and microbial communities in particular.

**Parameterisations for the competition between three grasses in salinity gradients.** The slim characterisation of the experimental setting reported in Ref. [34] (see Supplementary Notes) contrasts with the number of parameters of the DFTe functional $E[\mathbf{n}, \boldsymbol{\mu}](\mathbf{N})$ in Eq. (9), for which all components require input—except the environmental potential energies, which are constant in this uniform situation and therefore irrelevant. In order to proceed, we assume the following: (i) Coexistence of the three competing grasses in mixture suggests that three different resources are exclusively limiting each species; (ii) all species exhibit equal dispersal $\tau$ as well as interaction strength $\gamma$ factored into $\gamma_{ss'} = \gamma [f_s/f_{s'} - 1]_+$ (mixture- and monoculture abundances are not proportional, implying some kind of salinity-dependent competition); (iii) asymmetric interactions, since amensalism and repulsion yield distributions inconsistent with field data, see Supplementary Notes and Supplementary Fig. 5; (iv) as suggested in Ref. [34], we fix the salinity-dependent fitness proxies $f_s$ as the fraction of above-ground biomass of $s$ in mixture. This reflects increasing salt tolerance in the order Poa < Hord < Pucc. We then fit the nine parameters $(\tau, \sigma, \nu_{k \neq s}, \gamma)$ of

$$E(\mathbf{N}) = \frac{\tau}{2} \sum_{s=1}^{3} N_s^2 + \sum_{k=1}^{3} w_k \left[ \sum_{s=1}^{3} \nu_{ks} N_s - R_k \right]^2 + \sum_{\substack{s,s'=1 \\ s' \neq s}}^{S} N_s \gamma_{ss'} N_{s'} \quad (29)$$

to the abundances for the mixture in uniform salinity (see Supplementary Figs. 5 and 6), such that the nonuniform version of Eq. (29) allows us to predict the density distributions in Fig. 4 implied by heterogeneous resource distributions (see Supplementary Notes).

**Parameterisations for the dynamics of microbial predation.** *P. aurelia* (P) feeds on Cerophyl (c) and serves as the single resource for the predator *D. nasutum* (D). From the general DFTe energy we thus select the two quadratic resource energies and a parasitic interaction with fit parameter $\gamma$, akin to an asymmetric repulsive contact interaction that favours D. For this spatially uniform microbial system in suspension, $E[\mathbf{n}]$ reduces to the function

$$E(N_P, N_D) \equiv \mathcal{H}(\mathbf{N}) = \frac{A}{\rho_c^2} \left( \nu_{cP} \frac{N_P}{A} - \rho_c \right)^2 + \frac{A^3}{N_P^2} \left( \nu_{PD} \frac{N_D}{A} - \frac{N_P}{A} \right)^2 + \frac{\gamma}{A^2} N_D N_P^2. \quad (30)$$

We fit its minimiser $\hat{\mathbf{N}}$ (red cross in Fig. 5) to the average abundances (cyan cross in Fig. 5) for the last cycle of the most stable time series (Fig. 14c in ref. [36]); see also Supplementary Notes, Supplementary Fig. 9, and Supplementary Table 3. These average abundances anchor our comparison between the experimental data and the DFTe trajectory $\mathcal{H}(\hat{\mathbf{N}}) + \Delta E$, with a suitably chosen 'excitation' energy $\Delta E$. As expected, amensalistic interactions are not supported by the data and are therefore ruled out, see Supplementary Fig. 8 and Supplementary Fig. 10. By comparing, for example, the energies in Eqs. (29) and (30) for the grasses and the microbial predator–prey system, respectively, we see that DFTe is capable of determining what (very) different systems share—in this case the parasitic interaction, but not the dispersal pressure.

**Parameterisations for the complex large-scale synthetic community.** The community encompasses heterogeneous environments, competition over resources, all bipartite interactions of Supplementary Table 1, and predator–prey relations. From Eq. (9), we assemble

the according DFTe energy functional

$$E[\mathbf{n},\boldsymbol{\mu}](\mathbf{N}) = \int_A (\mathrm{d}\mathbf{r}) \sum_{s=1}^{S=7} \left( \frac{\tau_s}{2} n_s^2 + V_s^{\mathrm{env}} n_s + \sum_{(\alpha_s,\beta_{s'})} \sum_{\substack{s,s'=1 \\ s'\neq s}}^{S=7} n_s^{\alpha_s} \gamma_{ss'}^{(\alpha_s,\beta_{s'})} n_{s'}^{\beta_{s'}} \right)$$
$$+ \sum_{s=1}^{S=7} \mu_s \left( N_s - \int_A (\mathrm{d}\mathbf{r}) n_s \right) + \int_A (\mathrm{d}\mathbf{r}) \sum_{k=1}^{K=6} \left( \frac{1}{\rho_k^2} \sum_{s=1}^{S=7} \eta(\nu_{ks}) \exp\right.$$
$$\left[ \sigma\left(\frac{\lambda_{ks}}{\lambda_{ls}}-1\right)\right] \left) \left( \sum_{s'=1}^{S=7} \nu_{ks'} n_{s'} - \rho_k \right)^2 .$$

(31)

The species differ in their dispersal strengths $\tau_s$, habitat preferences $V_s^{\mathrm{env}}$, the types $(\alpha_s,\beta_{s'})$ of heterospecific interactions, modulated by interaction strengths $\gamma_{ss'}^{(\alpha_s,\beta_{s'})}$, as well as the types and amounts of required resources, see Supplementary Table 4. The synthetic environments and non-prey resource distributions are depicted in Supplementary Fig. 11. The density $n_s$ of each species may, independently from the other densities $n_{s'\neq s}$, refer to individuals per area, frequency, biomass density, fractional land cover, or any other expedient metric—units and absolute scales are absorbed in the parameters $\tau_s, V_s^{\mathrm{env}}$, and so forth. By deploying only contact-type interactions (see Supplementary Table 4), we implicitly state that members of any species are not directly influenced by conspecifics or heterospecifics beyond their local pixel of our coarse-grained area $A$. For example, if the Cat habitat area is ~50 km², then $A$ exceeds ~50,000 km² (see Supplementary Notes). This large-scale system contrasts with the fruit flies experiment in Fig. 2, where the long-range repulsive interaction between individuals turns out to couple all individuals explicitly across the entire area. Note that contact-type interactions on large spatial scales do not imply that the system is weakly interacting: Both the synthetic food web here and the fruit flies experiment are strongly interacting since the interaction energies comprise a substantial part of the total energy for both systems. Using the so specified energy functional along with the density expression of Eq. (4), we simulate $33 \times 33 = 1089$ parcels in the focal area $A = 1$. We obtain the equilibrium density profiles from a conjugate gradient descent towards the global energy minimum of Eq. (31) in the up to seven-dimensional space of abundances $\mathbf{N}$. In view of the stark disparity between species distributions (i) in isolation and (ii) under the influence of interactions, see Supplementary Fig. 14, we build intuition on the entire community by modelling subsystems with fewer species (Supplementary Figs. 12 and 13).

We illustrate the connections between DFTe inputs and outputs by explaining the small decrease of the Pig abundance upon Cat extermination (Fig. 6b): We observed that the Pig is the limiting resource for the Cat only in regions of very low Pig density (see Supplementary Fig. 13), such that changes in the Pig distribution are not primarily connected to the removal of the Cat. Second, the heavily predated Deer as well as the mutualistically connected Tree generally benefit from the absence of the Cat. As a result, the Grass and, consequently, the Snail come under pressure, which helps the Fungus, whose mutualistic connection to the Tree closes the positive feedback loop between Deer, Tree, and Fungus. The increased Fungus density then attracts the Pig, except in two small enclaves (indicated with arrows in Fig. 6b), where the processes just described are reversed, such that the Pig is mainly redistributed globally. Of course, this narrative conveys only the broad strokes of the quantitative simulation and plays out under the constraints imposed by environments and resources, but it conveys the quantitative knowledge about the community functioning obtained from the DFTe simulation.

**Parameterisations for the twenty tree species in a tropical forest.**
Since we lack additional ecological information beyond densities and

soil composition for constraining the full DFTe energy functional, we chose the same energy functional as in Eq. (31), though with (i) $S = 20$ species, which constitute more than 50% of the total basal area of the 328 censused species, (ii) all $K = 11$ measured resources, and (iii) the ansatz $V_s^{\mathrm{env}} = \epsilon_s^{\mathrm{alt}} V^{\mathrm{alt}} + \epsilon_s^{\mathrm{pH}} V^{\mathrm{pH}}$. Accordingly, there are 17 DFTe fit parameters per species—the two parameters $\epsilon_s^{\mathrm{alt}}$ and $\epsilon_s^{\mathrm{pH}}$ that encode the response to altitude and pH-level, eleven resource requirements $\nu_{ks}$ for $k = 1, ..., 11$, dispersal pressure $\tau_s$, and three fitness proxies $f_s^{(\mathrm{a/c})}, f_s^{(\mathrm{r/m})}$, and $f_s^{(\mathrm{asym})}$ that encode all the bipartite interactions of Supplementary Table 1 through the kernels $\gamma_{ss'}^{(\mathrm{i})} = f_s^{(\mathrm{i})} - f_{s'}^{(\mathrm{i})}$ for $s' > s$. We also leave $\sigma$ for the resource weights as a free parameter. All these 341 (281) DFTe parameters of the energy functional with (without) bipartite interactions are unknown and have to be extracted from fitting $E$ to the reference densities $\mathbf{n}^{\mathrm{ref}}$. This presents a high-dimensional constrained problem of maximising the least-squares overlap $\xi$ between the densities $\tilde{\mathbf{n}}(\bar{\mathbf{n}}^0)$ and $\mathbf{n}^{\mathrm{ref}}$, which we solve by using stochastic evolutionary algorithms that are efficient in escaping local optima, see Supplementary Notes.

### Reporting summary
Further information on research design is available in the Nature Portfolio Reporting Summary linked to this article.

### Data availability
The experimental data used in this manuscript are extracted from refs. [1,25,34,42,54]. The fruit-flies data from ref. [25] are available at https://doi.org/10.1038/s41467-018-05750-z. We digitised Fig. 6 of ref. [1], available at https://doi.org/10.2307/1937747, for displaying the experimental data on the competing algae in Supplementary Table 2. For our case study on three grasses, we obtained the above-ground biomass data in monoculture and mixture by digitising Fig. 3 of ref. [34], available at 10.1139/b91-310. For our predator–prey case study we used the digitised data of ref. [36], which is provided in the appendix of ref. [54], available at 10.1098/rspb.2000.1186. The data for Barro Colorado Island is accessible through Dryad at 10.15146/5xcp-0d46.

### Code availability
All the codes used in this study are available as open access repositories. The DFTe formalism for the heterogeneous systems is part of the C++ software package 'mpDPFT', available at https://doi.org/10.5281/zenodo.6999439. Analyses for the case studies of Figs. 2–7 are available as Mathematica notebooks at https://doi.org/10.5281/zenodo.7002198.

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

## Acknowledgements

We are grateful to James Patrick O'Dwyer for many insightful discussions and his crucial review of this work. We also thank Berthold-Georg Englert, Daniel Heyen, and Rosita Mei Ling Koh for valuable feedback on this manuscript. This research was supported by the ISF-NRF Singapore joint research programme (grant number WBS R-154-000-B09-281; R.A.C.).

## Author contributions

Both authors M.-I.T. and R.A.C. conceived the study and wrote the manuscript. M.-I.T. designed and carried out the research and analyses. R.A.C. conducted the Lotka–Volterra simulation and contributed to the choice of case studies for the research as well as to the interpretation of the analyses.

## Competing interests

The authors declare no competing interests.
