## [Peer Review File · Nature Communications]

Reviewer #1 (Remarks to the Author):

The authors aim to solve a long standing question in ecology that still remains poorly understood, namely, how we can scale from local processes such as species interactions or vital rates (mortality, reproduction) to large scale patterns such as species distribution and abundances. To answer this question the authors bring tools from physics, which are on the rise thanks to their scalability properties, and they want to apply them to the field of ecology.

The main aim of this study is rather interesting and I think it can be a timely contribution, yet I have a series of comments to the authors aiming to improve the core limitations (in my opinion) of this study.

1. If this paper wants to impact the broader field of ecology, I think the narrative should be improved, otherwise remains in many aspects too theoretical. In particular I have found the additional comments in the sup. material of the paper easier to understand than the main text in the results.

For instance, lines 71-73 "Here we construct the mechanistic DFTe energy E such that (i) its global minimum realises the ecosystem equilibrium with abundances N^* and (ii) its geometry away from N^* gives rise to both steady-state and non-equilibrium dynamics." I think it is easy to understand the first part in the sense that there should be a stable equilibrium, which is also feasible because species show positive abundances, but the second part remains more obscure. What type of geometry are the authors referring to? how it is possible that these deviations geometric deviations away from N^* lead to both equilibrium and non-equilibrium dynamics. Likewise, lines 82, what really means "an intra-specific pressure that implements one form of dispersal"? does this mean that dispersal kernels are species specific and depends and the shape depends on some density dependent mechanisms?. Again line 83, what means E does not introduce any quantum effects into ecology? this I can not understand.

Likewise, figure 1 is nice because aims to establish bridges between ecological concepts such as fitness differences or resource requirements with the methodology here presented such as DFTe hypersurface quantifies experimental steady-state cycle around equilibrium, but I feel that a better job could be done here. I overall suggest to create a box in which used concepts in this paper can be link with specific ideas brought by using the DFTe framework. In such regard, I can see that processes such as competition or predation is what the authors refer to species repulsion, but I wonder how we can interrelate the other ecological concepts.

2. The abstract overstate the idea that we need tools to better understand and predict the dynamics of large complex ecological systems that contain multiple species across large areas. However, the methodology here presented study low dimensional systems. Three out of the four databases are controlled experiments, and the maximum number of species analyze are four in the Tilman's experiment. Therefore there is a clear mismatch between the main goal of the study and the data analyzed. There are multiple data available such as these coming from Biotime dataset or communities from the Living Planet Index that the author can take and evaluate whether the methodology here proposed is suitable to describe observed patterns of changes in species abundances through time.

3. I can not evaluate how demanding in terms of data is this methodology. The authors provide links to the mathematica notebooks which I did not open honestly. But the point I want to make is that the data analyzed here is high resolution data, which is not the common pattern in ecology. What happens when there is missing data or sparse information. Moreover, I wonder to what extent the parameters used to fit the models were obtained from empirical data or they were arbitrarily taken from statistical distributions. For instance, the authors fixed the unit of length via $A=1$ in the Tilman's experiment. Similarly, the authors artificially reduced R^* values by a factor of 1000 (line 1055). I think these aspects which are key to decide whether this methodology

is useful needs to be addressed in detail.

So overall, I think the paper could impact the field of ecology if 1) it gains in clarity of how to link ecological concepts with the methodology presented, and 2) if it dissipates the doubts about the generality of the approach to be used in a wide range of context and not only in experimental systems with low species diversity, and quite importantly, 3) if it uses a less technical language that can be understood by a wide range of ecologists.

Reviewer #2 (Remarks to the Author):

see attachment.

Reviewer #3 (Remarks to the Author):

The authors introduce a standard method of many-body physics (density functional theory) to study multi-species communities.

After introducing the framework, the authors apply it successfully to 5 examples, modeling synthetic or empirical data.

I believe that there is an element of interest and value in the paper. Unfortunately, it does not emerge with clarity from the text. Here below a series of major and minor points.

Major point

1. My main concerns regard the role of DFT in being a tool that helps the modeling, the interpretation, and/or the conceptualization of ecological data. The output of DFT is equation 2, which is a self-consistent equation for the density of $n(r)$ as a function of the interactions with other species and the environment. DFT allows to obtain eq 2 from the general minimization of eq 9. A radical difference between many-body physics and ecology is that, in the former, the Hamiltonian is known from first principle. The problem in many-body physics is then mainly computational: given a Hamiltonian of a system compute the observables. DFT allows to reduce the many-body problem into an effective single-body problem (e.g. for the electron density) from which is possible to compute the interesting observables. In Ecology we do not have the "well-accepted" first principles from which we can write equations that describe the community dynamics (letting aside the fact that the equations that we typically write in ecology are not Hamiltonian). If we map the fixed points of the community dynamics into the minima of an effective (energy) function, there are not a-priori way to fix the form of the energy function. Contrary to many-body physics, the uncertainty in ecology does not come from the limited computational power that we have to solve the equations of motion, but from the uncertainty that we have around the equation of motions themselves.

2. The motivation behind the particular choice of the functional forms of the energy functions is unjustified. For instance, in equation 30, the interaction between *P. aurelia* and *D. nasutum* is modeled as a sum of quadratic forms, plus a term proportional to $N_D N_P^2$. Instead of raising the first two terms to the second power, one could have taken any convex function with a minimum in zero. Whether these alternatives, yet a-priori equivalent in terms of assumptions, choices would give the same results is unclear.

3. Related to the previous point, the usage of the word "mechanistic" is quite unclear in this context. The specific form of the Hamiltonian is chosen to be a sum of convex functions each with a minimum in the "right" position. There is no "mechanistic" interpretation of the shape of these functions. I am not arguing against "effective" models (any model is effective), but I am wondering of what the authors mean with "mechanistic" in the title.

4. The manuscript is very hard to follow. Here a few examples:

- the narrative of the introduction and first part of results is built around the parallel with how DFT is used in many-body physics, which make many part confusing for the audience without a background in physics (e.g. "the quantum-mechanical spin-statistics theorem" and the comparison between fermions and dispersal).
- other analogies are quite hard to parse. For instance: "quantum-gas-inspired mechanistic energy functional" is used when eq 1 is introduced. In what sense is "quantum-gas-inspired"? In what sense "mechanistic"? What is an "energy functional" in ecology?
- in eq 2, it appears $n_s[V_s - \mu_s](r)$. It is not explained what $[\]$ in this context? n_s is a functional of $V_s - \mu_s$ (I guess). But this is not explained when the equation appears.

Minor points

- "The resolution of such questions ... temporal and spatial scales", where "such questions" refer to the emergence of macro/mesoscopic properties from microscopic interactions. A philosophical comment / opinion, likely irrelevant. In stat mech, the emergence of the macroscopic description of reality (thermodynamics) is possible exactly because one restricts the analysis to a precise temporal (and spatial) scale. Thermodynamics work on the timescales longer than the equilibration times and shorter than whatever will impact the system in the future. And the general lesson of stat mech is that (at least close to a critical point, at least for critical exponent, which however are the main non-trivial macroscopic properties) microscopic details simply do not matter.
- line 69-71. It is unclear why (and in what sense) MaxEnt is equivalent to energy minimization.
- the description around eq 2 and 3 is quite obscure and does not clarify where the equation comes from and what V is. It is clarified in the methods, but it would be better to make it clear also in the text.
- lines 349-351. In what sense the conceptual foundation is "intuitive and straightforward"? How does it "unify the major ecological principles in a single mechanistic framework"?
- lines 353-355 + 360-361. Higher-order interactions (lines 360-361) is an example of where the components of the DFTe functional *cannot* be inferred from simpler subsystems. And it's unclear to me what is the difference with standard ecological theories (lines 351-353). If I believe in Lotka-Volterra equations, I can parameterize the equations using pairwise experiments and try to infer the outcome of the dynamics in larger communities. If it fails, one can change the model or include higher-order terms. Same for DFTe.

Response to Referee 1:

All page and line numbers, references, and equations mentioned in the authors' responses refer to the original manuscript (submitted on June 16, 2021), unless stated otherwise. Grey-shaded boxes are reviewer comments. Unshaded boxes with black frame are our amendments of the original manuscript. Text outside of boxes is directed at the reviewer. Some of our responses are in boxes with green frame, labelled with 'Author Comment' (clickable), and referred to in other responses, see page 1 for an overview.

Reviewer 1 (Remarks to the Author):

The authors aim to solve a long standing question in ecology that still remains poorly understood, namely, how we can scale from local processes such as species interactions or vital rates (mortality, reproduction) to large scale patterns such as species distribution and abundances. To answer this question the authors bring tools from physics, which are on the rise thanks to due their scalability properties, and they want to apply them to the field of ecology.

The main aim of this study is rather interesting and I think it can be a timely contribution, yet I have a series of comments to the authors aiming to improve the core limitations (in my opinion) of this study.

We thank the Referee for the kind assessment that our work is 'interesting and timely' and are grateful for the Referee's advice on how to improve the study in its core messages—we hope that the amendments of our manuscript and our accompanying responses to the remarks of all three Referees are to the point.

1. If this paper wants to impact the broader field of ecology, I think the narrative should be improved, otherwise remains in many aspects too theoretical. In particular I have found the additional comments in the sup. material of the paper easier to understand than the main text in the results.

For instance, lines 71-73 'Here we construct the mechanistic DFTe energy E such that (i) its global minimum realises the ecosystem equilibrium with abundances \hat{N} and (ii) its geometry away from \hat{N} gives rise to both steady-state and non-equilibrium dynamics.' I think it is easy to understand the first part in the sense that there should be a stable equilibrium, which is also feasible because species show positive abundances, but the second part remains more obscure. What type of geometry are the authors referring to? how it is possible that these deviations geometric deviations away from \hat{N} lead to both equilibrium and non-equilibrium dynamics.

Author Comment 2. We have improved the narrative throughout the manuscript, while paying special attention (i) to the core ingredients of the DFTe formalism that are introduced on pages 3/4 and (ii) to the clarity of technical aspects, as detailed in the following as well as in Author Comments 3 and 8.

We completely overhauled the narrative and phrasing around the four case studies illustrated in Figs. 2–5, on lines 132–254 (lines 175–311 in the revised manuscript), where key aspects of DFTe are discussed. We also completely overhauled the Discussion on lines 333–388 (lines 406–483 in the revised manuscript). All other amendments to our original manuscript are spelled out explicitly in the present document.

We replaced the sentence on lines 71–73 [and so hint at our explanations of the results shown in Fig. 5 and at the DFTE hypersurface introduced in equation (14)] by:

Here, we construct the mechanistic DFTE energy $E(\mathbf{N})$ such that its global minimum realises the ecosystem equilibrium with abundances $\mathbf{N} = \hat{\mathbf{N}}$. While we focussed on establishing the equilibrium properties of DFTE as a necessary step towards the modelling of dynamics, we note that E represents an S –dimensional (hyper-)surface whose shape can determine (i) steady-state dynamics (see Methods)—in analogy to, for example, Kepler orbits in a gravitational field—and (ii) nonequilibrium dynamics, where also drivers such as stochastic environments can prevent equilibration.

On line 401 we added

We may illustrate the effect of equation (5) for an amphibian species s by assigning values $V_s^{\text{env}} < 0$ in a lake, while $V_s^{\text{env}} > 0$ in the surrounding desert penalises high densities n_s as we minimise the total energy that includes $E_{\text{env}}[n_s]$.

On line 428 we added

If equation (7) is the total energy functional, then a single-species system with a single resource equilibrates with density $n_1(\mathbf{r}) = \rho_1(\mathbf{r})/\nu_{11}$ at every position \mathbf{r} , and additional DFTE energy components would modify this equilibrium.

Likewise, lines 82, what really means ‘an intra-specific pressure that implements one form of dispersal’? does this mean that dispersal kernels are species specific and depends and the shape depends on some density dependent mechanisms?.

Indeed, by ‘an intra-specific pressure that implements one form of dispersal’ on line 81, we mean that dispersal kernels are species-specific and that their shape depends on some density-dependent mechanisms. Since we had declared these properties only implicitly on lines 391–398, we added on line 398

The functional in equation (4) is species-specific due to τ_s , and we assume that the quadratic density dependence is the result of a microscopic (viz., individual-based) mechanism that is not necessarily known to us.

Again line 83, what means E does not introduce any quantum effects into ecology? this I can not understand.

Author Comment 3. With the sentence ‘But this declaration of E does not introduce any quantum effects into ecology.’ on line 82 we merely meant to assure (potentially concerned) readers that we do not introduce any quantum effects into ecology despite the fact that the mathematical structure of DFTE is borrowed in part from quantum physics. Since we now believe that our referrals to quantum physics confuse more than they clarify, we amend most statements regarding quantum physics.

We replaced the sentence on lines 51 & 52 by

The core of DFTE is an energy functional for ecosystems that merges ecological principles with insights from DFT. Though DFTE draws inspiration in particular from analogies with the TF model of multi-component quantum gases, it does not of course introduce any quantum effects into ecology itself.

Moving the content from lines 391–401, we replaced ‘We base the additive [...] energetic quantities and

mechanisms.’ on lines 76–87 by

We base the additive decomposition of E on our understanding of how constituents of physical systems come together to assemble the energy and determine the functional form of E by introducing and building on intuitive concepts in analogy to the energy components of physical systems. For example, for the dispersal energy E_{dis} we adopt the TF kinetic energy expression for species s :

$$E_{\text{dis}}[n_s] = \frac{\tau_s}{2} \int_A (d\mathbf{r}) n_s(\mathbf{r})^2, \quad (4)$$

where τ_s is a species-specific dispersal pressure constant, and the integral is over all spatial positions \mathbf{r} in a focal area A . Equation (4) encodes an intra-specific pressure of species s and implements one form of dispersal where, for example, individuals repel conspecifics from territory; note that positive (negative) energy density encodes repulsion (attraction). The functional form E_{dis} in equation (4) implies a conspecific negative density dependence, relative to other energy components whose integrands scale less than quadratically with density. We emphasise that we neither aim at describing all forms of dispersal via equation (4) nor necessarily need to include E_{dis} even if a specific ecosystem exhibits dispersal, which can potentially be modelled with other energy components. The second term in equation (1) is the environmental energy, which for each species s we model via analogy to the external energy in physics:

$$E_{\text{env}}[n_s] = \int_A (d\mathbf{r}) V_s^{\text{env}}(\mathbf{r}) n_s(\mathbf{r}), \quad (5)$$

where $V_s^{\text{env}}(\mathbf{r})$ governs the effect of the environment on the energy for species s . Equation (5) parallels the parameterisation in Ref. [25] and models, for example, habitat preference and external influences, such as spatially varying climate and local deforestation stress. Summing equations (4) and (5) over s , we obtain the respective total energy components displayed in equation (1).

On line 87 we add

We emphasise that we do not pursue the hopeless goal of establishing DFTe based on actual chemical quantities and mechanisms.

After ‘to equation (1).’ on line 391, we added

The methodology of DFTe can be understood by inspecting the dispersal and environmental energies in equations (4) and (5) without interactions.

and we replaced ‘Inspired by analogues of interacting quantum gases, we let’ on line 406 by

In the following we make explicit the interaction and resource energies that enter equation (1) and are used in our case studies of Figs. 2–7. We let

We replaced the sentence on lines 401–403 by

In our first case study, illustrated in Fig. 2 and Extended Data Fig. 1, we demonstrate that equation (5), in conjunction with equation (4), can realistically describe the influence of the environment on species' distributions. Mechanisms that alter the trade-off between dispersal and environment can be introduced as part of E_{int} . For instance, back reactions on the environment could be modelled with a bifunctional [...]

and moved the sentence on lines 405–406 to line 87.

We replaced lines 63–64 by

We have transferred the DFT narrative and methodology to ecology by building a cost function, the mechanistic energy functional [...]

We replaced the sentence on lines 414–416 by

The mechanistic effect of these interaction kernels on the density distributions is the same in ecology as it is in physics—a mathematical insight that inspired us to build ecological analogues to the phenomenology of quantum gases, which feature functionals of the kind in equation (6). Note that we do not introduce any quantum effects into ecology despite the fact that the mathematical structure of DFTE is borrowed in part from quantum physics.

On line 418 we added

Both types of interactions feature in describing the ecosystems addressed in this work.

Likewise, figure 1 is nice because aims to establish bridges between ecological concepts such as fitness differences or resource requirements with the methodology here presents such as DFTE hypersurface quantifies experimental steady-state cycle around equilibrium, but I feel that a better job could be done here. I overall suggest to create a box in which used concepts in this paper can be link with specific ideas brought by using the DFTE framework. In such regard, I can see that processes such as competition or predation is what the authors refer to species repulsion, but I wonder how we can interrelate the other ecological concepts.

We thank the Referee for compelling us to better describe the links between ecological concepts and the DFTE framework. This is not only essential to gain an intuition for DFTE and to apply it in practice. We believe that the interpretations of explicit DFTE energy functionals and the intuition behind their construction *constitutes* these links. That said, we intended Fig. 1 to be a summary of the manuscript and in particular of the results extracted from DFTE. Figure 1 displays the links between ecological concepts and components of the DFTE framework only implicitly/superficially—also because a real understanding of these links has to come from interpreting the mathematical structure of the four modular components of the DFTE energy, which are listed in Fig. 1a and explained in Methods; see also Extended Data Table 1, where ecological concepts are linked explicitly with those DFTE energy components that are specified by the interaction kernels $\gamma_{ss'}$. The following amendments are intended to better link ecological concepts with the DFTE framework, especially in Results.

We added the explicit links (beyond inter-species interactions) between ecological concepts/phenomena and specific DFTE components to Extended Data Tab. 1

intra-specific interaction	τ_s		interpretation
spreading/dispersal	> 0		avoidance of conspecifics
aggregation	< 0		beneficial proximity of conspecifics
environment	V_s^{env}		interpretation
attraction	small		favourable region (a type of niche)
repulsion	large		hostile region
bipartite interactions	$\gamma_{ss'}$	$(\alpha_s, \beta_{s'})$	impact on (s, s')
amensalism	> 0	(0,1)	(0,-)
repulsion	> 0	(1,1)	(-,-)
asymmetric interaction (e.g., parasitism)	> 0	(1,2)	(+,-)
commensalism	< 0	(0,1)	(0,+)
mutualism	< 0	(1,1)	(+,+)

and added the following to the caption (now titled ‘Links between ecological concepts/phenomena and DFTE parameters.’) on line 703:

We summarise and complement the interpretation of the DFTE parameters introduced through equations (4)–(6).

We overhauled Fig. 1, in particular panel a, to direct the reader to the relevant explanations of aforementioned links:

On lines 123–124 we amended accordingly ‘[...] **b–g**, We highlight our main results for six experimental and synthetic systems (Figs. 2–7) [...]’ and replaced ‘five’ by ‘six’ on lines 109 & 116.

After line 117, we added

We note that specific models constructed within the DFTE framework do not necessarily encode microscopic causal relations, but may provide effective descriptions, where ecological phenomena cannot always be linked unambiguously to specific DFTE components. The corresponding links declared in Extended Data Tab. 1 are therefore prototypical rather than rigorous. For example, in our first case study, we may attribute the territoriality of fruit flies either to the local dispersal pressure or to the nonlocal repulsion or some combination of the two (Fig. 1b).

On line 883, we added another illustration of the intuition behind building energy functionals in DFTE—in this case for the ecological phenomenon of bet hedging, see Author Comment 14.

2. The abstract overstate the idea that we need tools to better understand and predict the dynamics of large complex ecological systems that contain multiple species across large areas. However, the methodology here presented study low dimensional systems. Three out of the four databases are controlled experiments, and the maximum number of species analyze are four in the Tilman's experiment. Therefore there is a clear mismatch between the main goal of the study and the data analyzed. There are multiple data available such as these coming from Biotime dataset or communities from the Living Planet Index that the author can take and evaluate whether the methodology here proposed is suitable to describe observed patterns of changes in species abundances through time.

Indeed, we emphasised the need for new holistic descriptions of ecosystems across scales. On lines 21–30 we had cited a selection [3,4,8–12] of the extensive body of literature that supports the corresponding statements in the abstract. We proposed DFTE as a general framework and speculated that it will be useful in future, while pointing to various challenges of modelling large complex ecosystems that we do not overcome in the current manuscript (see lines 125–127, 340–342, and 364–382). Also, on lines 379–382 we point to a possible agenda for studying dynamics within DFTE, which is outlined on pages 59–62. Although we do discuss dynamics with the aid of the predator–prey cycle in Fig. 5 and with the restricted trajectory in Extended Data Fig. 2, the current manuscript focuses on equilibrium properties, not on explaining field observations of changes in species abundances through time. We amended lines 71–73 to make this point more explicit, see Author Comment 2.

The synthetic seven-species example of Fig. 6 may indeed be viewed as an inadequate substitute for actual ecosystems. Although it allowed us to demonstrate how DFTE can be fruitfully applied to high-dimensional real-world data *in principle*, we agree with the Referee that we had not demonstrated the applicability of DFTE to such many-species real-world data *in practice*.

Author Comment 4. To address the Referee's absolutely valid question of scalability towards many-species communities, we used DFTE to model the twenty most abundant tree species observed in one of the world's most studied sites, the 50ha-plot on Barro Colorado Island, Panama. In fact, we used this example to simultaneously address several concerns and questions raised by the Referees, including (i) scalability of DFTE, (ii) how to deal with sparse information, (iii) how to distil general properties of ecosystems, (iv) how to efficiently minimise high-dimensional functions with many local minima, and (v) what the freedom in choosing DFTE energy components can mean in practice.

After line 331, we added

Finally, we explored the scalability of DFTE towards multi-species systems by modelling the twenty most abundant tree species found in the tropical forest plot on Barro Colorado Island (BCI), Panama [Condit, R. *et al.* Complete data from the Barro Colorado 50-ha plot: 423617 trees, 35 years, *Dryad, Dataset*, (2019)]. The mechanisms underlying the diversity of tropical forests in general, and the BCI plot in particular, are still not yet fully understood, but have broad implications for ecological theory and policies aimed at maintaining biodiversity [Davies, S. J. *et al.* ForestGEO: Understanding forest diversity and dynamics through a global observatory network. *Biol. Conserv.* **253**, 108907 (2021); Wiegand, T. *et al.* Consequences of spatial patterns for coexistence in species-rich plant communities, *Nat. Ecol. Evol.* **5**, 965 (2021)]. Our BCI case study shows how DFTE can deal with an extreme case of missing information, since we use no data beyond the observed data on abundances, soil, and altitude (see Methods). This is not a situation most suited for DFTE models, which are preferably built by determining some parameters from simpler versions of the ecosystem. Here, for example, if the resource requirements were known from monoculture experiments, we could eliminate 220 out of up to 341 fit parameters. High variability arises in the fitted parameters across simulation runs because the large number of energy components of E permits an increase in some components and a decrease in others without significant impact on the density distributions. We can nonetheless draw some general conclusions: the DFTE simulations with and without bipartite interactions yield spatial basal area distributions for each species of similar quality (Fig. 7a–c) despite disparate species-specific fit parameters (Fig. 7d). These results support the hypothesis that inter-species interactions in many-species communities are weak relative to other mechanisms [Hubbell, S. P. & Foster, R. B. Biology, Chance and History and the Structure of Tropical Rain Forest Tree Communities, pp. 314–329 in *Community Ecology*, Diamond, J.M. & Case, T. J. (Eds.), Harper and Row, New York (1986)]. In the DFTE fit to BCI, the total interaction energy is relatively small and the extracted interaction kernels (Fig. 7e) are distributed across positive and negative values for all bipartite interactions. Similar conclusions have been reached in previous studies using a maximum entropy approach [Volkov, I. *et al.* Inferring species interactions in tropical forests, *PNAS* **106**, 13854 (2009)], a stochastic birth–death model [Volkov, I. *et al.* Inferring species interactions in tropical forests, *PNAS* **106**, 13854 (2009)], spatial point pattern analysis [Wiegand, T. *et al.* Testing the independent species’ arrangement assertion made by theories of stochastic geometry of biodiversity, *Proc. R. Soc. B: Biol. Sci.* **279**, 3312 (2012)], and a specialised multi-scale analysis [Wiegand, T. *et al.* Consequences of spatial patterns for coexistence in species-rich plant communities, *Nat. Ecol. Evol.* **5**, 965 (2021)].

FIG. 7. An application to the tropical-forest data from Barro Colorado Island demonstrates the scalability of DFTE towards many-species systems. **a**, We obtained the DFTE densities \tilde{n}_s (\tilde{n}_s^0), which represent the spatially resolved fractional basal area of species s averaged over all available censuses [Condit2019], in the top (centre) row of charts by simultaneously fitting the general DFTE energy functional E with (without) inter-species interactions to the reference densities n_s^{ref} in the bottom charts; see Methods. Both \tilde{n}_s and \tilde{n}_s^0 are an adequate fit ($\xi \approx 0.79$) to n_s^{ref} for all twenty species. In the three charts for each of the species shown ($s = 1, 4, 8, 12, 16, 20$) we report the density values relative to $\max_r \{ \tilde{n}_s(\mathbf{r}), \tilde{n}_s^0(\mathbf{r}), n_s^{\text{ref}}(\mathbf{r}) \}$. Complementing measures of the quality of \tilde{n} are the species-resolved variability in density (**b**) and the (sorted) least-squares overlaps ξ of abundances $N(Q)$ with $N^{\text{ref}}(Q)$, viz. the overlaps of local species-abundance distributions (SAD) for each of the 45 quadrats Q (**c**). **d**, The fit parameters associated with altitude, pH-level, and dispersal, with and without bipartite interactions. **e**, The fit results for the bipartite interaction kernels (see Methods); each sub-panel is normalised to its respective maximum value.

After line 582, we added

Figure 7. Since we lack additional ecological information beyond densities and soil composition for constraining the full DFTE energy functional, we chose the same energy functional as in equation (31), though with (i) $S = 20$ species, which constitute more than 50% of the total basal area of the 328 censused species, (ii) all $K = 11$ measured resources, and (iii) the ansatz $V_s^{\text{env}} = \epsilon_s^{\text{alt}} V^{\text{alt}} + \epsilon_s^{\text{pH}} V^{\text{pH}}$. Accordingly, there are 17 DFTE fit parameters per species—the two parameters ϵ_s^{alt} and ϵ_s^{pH} that encode the response to altitude and pH-level, eleven resource requirements v_{ks} for $k = 1, \dots, 11$, dispersal pressure τ_s , and three fitness proxies $f_s^{(a/c)}$, $f_s^{(t/m)}$, and $f_s^{(asym)}$ that encode all the bipartite interactions of Extended Data Tab. 1 through the kernels $\gamma_{ss'}^{(i)} = f_s^{(i)} - f_{s'}^{(i)}$ for $s' > s$. We also leave σ for the resource weights as a free parameter. All these 341 (281) DFTE parameters of the energy functional with (without) bipartite interactions are unknown and have to be extracted from fitting E to the reference densities n^{ref} . This presents a high-dimensional constrained problem of maximising the least-squares overlap ξ between the densities \tilde{n} (\tilde{n}^0) and n^{ref} , which we solve by using stochastic evolutionary algorithms that are efficient in escaping local optima, see Supplementary Notes.

After line 1384, we added

Notes on Fig. 7. We obtained the reference density data shown in Fig. 7 and Supplementary Fig. 8. as follows. While we considered all data from the eight censuses carried out between 1982 and 2015, we omitted about 700 (out of more than 400000) entries of the raw data where consistency checks (same tree-ID & same tag & same coordinates) for individual trees failed. After calculating the average basal area $\langle B(t) \rangle$ of each individual tree t across the eight censuses, we assigned the abundance $N_s(q)$ of a species s in quadrat q (with area $20 \text{ m} \times 20 \text{ m}$) of the fifty-hectare plot by accumulating all $\langle B(t) \rangle$ that belong to a tree of species s located within q and so identify the $S = 20$ most abundant species in order of decreasing abundance from $s = 1$ (*Quararibea asterolepis*) to $s = 20$ (*Virola sebifera*), see Supplementary Tab. 2. We interpolated the measurements of the two environmental variables of altitude and pH-level as well as the eleven resources (Al, B, Ca, Cu, Fe, K, Mg, Mn, P, Zn, N) to obtain their values at the centres of the quadrats q . By averaging over multiple quadrats q , we obtained the environmental base potentials $V^{\text{alt}}(\mathbf{r})$ and $V^{\text{pH}}(\mathbf{r})$ as well as the resource densities $\rho_k(\mathbf{r})$ for $k = 1, \dots, 11$ with centre coordinates \mathbf{r} of $9 \times 5 = 45$ larger quadrats Q of the $1000 \text{ m} \times 500 \text{ m}$ plot. This coarse-graining tempers the effects of stochastic processes on local densities and reduces computational cost. The bottom bar charts in Fig. 7a and in Supplementary Fig. 8. show the resulting reference densities $n_s^{\text{ref}}(\mathbf{r})$ for $s = 1, \dots, 20$.

We employed two stochastic optimisers to obtain the fits to \mathbf{n}^{ref} , a particle swarm optimisation (PSO) and a genetic algorithm (GA). Both can optimise non-convex functions with many local optima in high-dimensional spaces, but we found our implementation of PSO to yield better results more quickly than the GA that we had adapted from the openGA library [Mohammadi, A. *et al.* OpenGA, a C++ Genetic Algorithm Library, *IEEE International Conference on Systems, Man, and Cybernetics*, 2051 (2017)]. We optimised in the initially large $(17 \times S)$ -dimensional ($(14 \times S)$ -dimensional in the non-interacting case) search space by (i) expanding and shrinking individual dimensions adaptively and (ii) re-initialising the search in 20 subsequent runs each inheriting the adapted search space of the previous run. Through testing on few-species subsystems of the BCI data, we observed that this procedure provides a satisfactory trade-off between computing time and fit quality. Fixing all parameters except the three interaction parameters for each species to their fitted values, and further optimising in the then much reduced parameter space, we only found marginally better fits to the density data, which suggests that the optimisation had already converged to a good local optimum.

Supplementary Fig. 8. **DFTe fit results and BCI reference data.** As in Fig. 7, but for the remaining fourteen species $s = 2, 3, 5-7, 9-11, 13-15, 17-19$, see Supplementary Tab. 2: We obtained the DFTe densities \tilde{n}_s (\tilde{n}_s^0) in the top (centre) charts, which represent the average fractional basal area over the eight censuses of basal area since 1982 [Condit2019], from fitting the full DFTe energy functional with (without) interspecies interactions to the reference densities n_s^{ref} in the bottom charts. The density values shown in the three bar charts for each species s are rescaled to the unit interval via division by the maximum of \tilde{n}_s , \tilde{n}_s^0 , and n_s^{ref} .

species index s	species name	ForestGEO identifier	N^{ref} [m ²]	ξ	ξ^0
1	Quararibea asterolepis	QUARAS	108.3	0.791	0.789
2	Trichilia tuberculata	TRI2TU	95.7	0.792	0.790
3	Alseis blackiana	ALSEBL	69.5	0.791	0.793
4	Hura crepitans	HURACR	65.7	0.791	0.789
5	Prioria copaifera	PRI2CO	64.7	0.791	0.788
6	Ceiba pentandra	CEIBPE	59.8	0.791	0.789
7	Faramea occidentalis	FARAO	57.4	0.795	0.791
8	Jacaranda copaia	JAC1CO	38.5	0.791	0.792
9	Anacardium excelsum	ANACEX	32.8	0.801	0.794
10	Apeiba membranacea	APEIME	32.6	0.792	0.789
11	Brosimum alicastrum	BROSAL	30.7	0.792	0.803
12	Beilschmiedia pendula	BEILPE	30.6	0.791	0.788
13	Tetragastris panamensis	TET2PA	30.3	0.793	0.807
14	Poulsenia armata	POULAR	29.1	0.793	0.791
15	Tabernaemontana arborea	TAB2AR	28.4	0.793	0.788
16	Cavanillesia platanifolia	CAVAPL	24.7	0.793	0.793
17	Virola surinamensis	VIROSU	24.6	0.791	0.789
18	Hirtella triandra	HIRTTR	22.5	0.794	0.790
19	Ocotea whitei	OCOTWH	22.2	0.793	0.788
20	Virola sebifera	VIROSE	21.0	0.794	0.797

Supplementary Tab. 2. **Tree species modelled in the case study of Fig. 7.** The twenty most abundant species in terms of the average total basal areas N^{ref} , as extracted from the eight censuses of the 50-ha plot on Barro Colorado Island [Condit2019]. We maximised the overlap ξ (ξ^0) between abundance data and DFTE simulation with (without) bipartite interactions by maximising the smallest overlap, here incidentally realised by $s = 4$ ($s = 5$), among all twenty species.

3. I can not evaluate how demanding in terms of data is this methodology. The authors provide links to the mathematica notebooks which I did not open honestly. But the point I want to make is that the data analyzed here is high resolution data, which is not the common pattern in ecology. What happens when there is missing data or sparse information.

Indeed, DFT does not handle missing data in an unbiased way as statistical methods like MaxEnt do. With our application to three grasses in a saline environment, we showed how DFTE can project sparse information from uniform experimental setups to density distributions in heterogeneous environments by making the unknown resource requirements fitted parameters. We hope to have clarified this aspect with the rephrased narrative of this case study on lines 253–297 in the revised manuscript. We also added (at the end of ‘Notes on the DFTE energy functional.’ in the Supplementary Notes)

We note that when fitting the DFTE energy to scattered or low-resolution data, the continuum between the data points (of, e.g., V^{env}) has to be interpolated, and the resulting predictions will reflect the quality of the chosen interpolation. In general, the number of free DFTE parameters increases with the sparsity of information, which is a challenge also for other mechanistic approaches.

Our analysis of the BCI data in Author Comment 4 is an extreme example of that sort. There, all DFTE parameters are unknown and are extracted from fitting E to the available density data.

Moreover, I wonder to what extent the parameters used to fit the models were obtained from empirical data or they were arbitrarily taken from statistical distributions.

Author Comment 5. No parameters of the specific data-driven models, that is, equations (24) and (28)–(30), were taken arbitrarily—and none from any statistical distribution. The parameters are either fit parameters or are inferred from the data, as described on lines 502–582. We would be grateful if the Referee could point us to parts of the manuscript where readers might get a different impression. After line 502, we added

We also note that all DFTE parameters of the specific data-driven models pertinent to Figs. 2–5 and 7 are either fit parameters or are inferred from the data.

In this regard, we had conceded on lines 366–372 that a (rudimentary) intuition is required to connect data with an appropriate DFTE energy component and its parameters. The food web of Fig. 6 is entirely synthetic, and we added on line 275:

We selected the parameters of this food web solely for pedagogical reasons, namely to reveal the functioning of all ingredients of the DFTE energy in a controlled way.

For instance, the authors fixed the unit of length via $A=1$ in the Tilman’s experiment.

Tilman’s experiment presents a uniform situation with local (contact-type, so we presume) interactions. That is, the length scale \sqrt{A} can be set arbitrarily. Its only purpose is to declare the unit of length (for completeness), but is of no consequence for determining the equilibrium abundances; see lines 717–720 and 866–867 for additional comments on units. We replaced the sentence on lines 1036–1037 by

The uniform setup with local interactions allows us to set the unit of length arbitrarily, without consequences for the determination of the equilibrium abundances. We set $A = 1$, which amounts to 1 mL suspension by our definition.

Similarly, the authors artificially reduced R^* values by a factor of 1000 (line 1055). I think these aspects which are key to decide whether this methodology is useful needs to be addressed in detail.

We replaced ‘(over-)consumption of magnitude R^* [...] for \mathcal{R}_6 .’ on lines 1054–1055 by

(over-)consumption of magnitude R^* : If we artificially reduce the R^* -values by a factor of 1000, R^* -theory indeed predicts $(F|A|S) = (0|72|170)$ for \mathcal{R}_6 , which is almost exactly the DFTE prediction in Extended Data Tab. 2. Note that we reduce the R^* -values only here, and for the sole purpose of demonstrating that our choice of the DFTE energy in equation (28) for Tilman’s experiment is appropriate if the supplied resources are not too close to the R^* -values. We used the unchanged R^* -values from Ref. [1] for all outcomes presented in Results and in Extended Data Tab. 2.

So overall, I think the paper could impact the field of ecology if 1) it gains in clarity of how to link ecological concepts with the methodology presented, and 2) if it dissipates the doubts about the generality of the approach to be used in a wide range of context and not only in experimental systems with low species diversity, and quite importantly, 3) if it uses a less technical language that can be understood by a wide range of ecologists.

We hope that the Referee will find the above revisions adequate—together with our amendments in response to the editor's and to the other referees' comments.

Response to Referee 2:

All page and line numbers, references, and equations mentioned in the authors' responses refer to the original manuscript (submitted on June 16, 2021), unless stated otherwise. Grey-shaded boxes are reviewer comments. Unshaded boxes with black frame are our amendments of the original manuscript. Text outside of boxes is directed at the reviewer. Some of our responses are in boxes with green frame, labelled with 'Author Comment' (clickable), and referred to in other responses, see page 1 for an overview.

Reviewer 2 (Remarks to the Author):

The authors here draw on density functional theory, a powerful tool borrowed from physics, in order to implement an ecologically-informed version of the theory. The approach is novel and interesting, but I think that the authors should tone down some of their claims and be more cautious in interpreting their results.

We are happy to hear that our work is 'novel and interesting' and are grateful for the Referee's advice to take a more balanced stance on the reach of our results and interpretations—we hope that the amendments of our manuscript and our accompanying responses to the remarks of all three Referees are to the point.

We completely overhauled the narrative and phrasing around the four case studies illustrated in Figs. 2–5, on lines 132–254 (lines 175–311 in the revised manuscript), where key aspects of DFTe are discussed. We also completely overhauled the Discussion on lines 333–388 (lines 406–483 in the revised manuscript). All other amendments to our original manuscript are spelled out explicitly in the present document.

General comments.

I find this tool very useful for modelling or making predictions about specific instances of ecosystems, when the phenomenological guess of the ecological processes is sufficient to produce at least good qualitative results. However, I am not sure about its potential when we have to single out the main drivers of ecosystems, or how to distil general or fundamental properties/ processes, or understanding scaling relations in empirical patterns across very different spatial or temporal scales.

For example, it may well be that different energy functionals provide similar profiles for density distributions when we find multiple minima that are very close to each other or one minimum that is very shallow. This means that one cannot discriminate between, e.g., a new free parameter within a known component of the total energy functional, or a brand new component of the energy functional — which accounts for a different mechanism — which contains only one free parameter. To a certain extent, this framework looks hardly falsifiable, because one can keep adding new components to the energy functional to fit the empirical distributions. However, adding or modifying new terms does not mean that we are unifying ecological processes nor that we are understanding better the system.

Author Comment 6. We agree with the Referee that adding more and more energy terms to align predictions with data or to compensate for missing data is no unification. We made a related declaration on lines 364–366, but believe that our usage of 'unify' contributed to the Referee's concerns. We never intended to give the impression that DFTe unifies ecological phenomena or principles by describing them through the same or similar mathematical formulae. We therefore replaced 'unifies ecology' on line 14 by 'models ecological phenomena' and removed the sentence on line 349–351 (with part of its content included in the completely overhauled Discussion).

We also replaced ‘Despite numerous attempts, no definitive unification that bridges these theories across scales has yet been achieved. Ideally, a unified theory’ on lines 24–25 by

Nevertheless, theories bridging different scales are lacking. Ideally, models

We moved lines 1073–1080 on the ambiguity of the energy functional to after line 542:

It would be surprising if equation (25) were the only possible choice for the interaction kernel, and it is quite likely that many different energy functionals that encode different mechanisms and input parameters are equally suited for reproducing the data targeted in this example. Only future DFTE studies that take into account additional data can reduce the space of possible functionals. Taking the cue from physics, we may hope that a universal functional will prevail and thereby provide deeper insights into the fundamental mechanisms and relations underlying ecological systems in general and microbial communities in particular.

After line 894, we added (see also Author Comment 9)

The DFTE framework is not falsifiable since additional energy components can be added to equation (1), but particular DFTE models created for specific ecosystems can be ruled out with confidence. This would be of little use if the models would comprise many energy functional components with many fit parameters and if new phenomena needed new functional forms. However, assuming that our case studies in Figs. 2–7 are representative, we have demonstrated that our four energy functionals are able to cover the basic ecosystem properties across scales, and that the parametric variability in these functionals allows us to distil general ecosystem properties. For example, DFTE fits the fruit flies data with a total of two parameters from three energy components, and we would obtain an inferior fit if we replaced the repulsive Coulomb-type interaction by an attractive interaction. This way, we have distilled that long-range repulsive behaviour rather than attraction is a fundamental property of fruit flies. Of course, specific DFTE models select a prediction from a continuum of possibilities, and do not necessarily answer yes–no questions.

In this manuscript, we have left many important aspects of ecosystem modelling untouched. For example, we have so far modelled systems only at fixed scales, but we speculate that the understanding of scaling relations across scales within the DFTE framework could come from modelling a fixed set of species with a fixed functional at different scales. If the fits to the data are of similar (and high enough) quality across these scales, we may argue that the mechanisms underlying the employed energy functional are responsible for the scaling relations we observe in the data.

The mechanistic nature of the processes in the DFTE is phenomenological and not emergent from microscopic interactions.

Author Comment 7.

We agree and note that we mentioned nowhere the emerging of any DFTE ingredients from microscopic interactions. We had declared on lines 203–206 that DFTE is an effective description and contrasts with approaches that rely exclusively on microscopic detail. We overhauled the Discussion completely, also to clarify the phenomenological and mechanistic character of DFTE. See also Author Comment 8 on the existence of DFTE potentials.

To some degree, the approach looks similar to what has been done in the past in other fields, e.g., classical mechanics applied to economic theory, gravitation-like models applied to human mobility... this latter produced the gravity model, now superseded by the radiation model — far less phenomenological than the former — which basically has no free parameters.

We sincerely thank the Referee for pointing us to this fitting parallel in model development. We added an according reference on lines 474–476 in the revised manuscript:

History provides vivid examples of physics-inspired models that have led to powerful frameworks in other fields, such as the gravity model of human mobility, whose initial heuristic foundations have been refined over decades towards a parameter-free model [Simini, F. *et al.* A universal model for mobility and migration patterns, *Nature* **484**, 96 (2012)].

Along this way, the DFTe needs potentials which in ecological systems not only are not measurable, but sometimes do not even exist. There is no compelling reason why in ecology we should expect that energies can be derived from potentials, but this is what the authors assume when using, e.g., eq.3 or eq.5...

Author Comment 8. We added the following clarification after line 502:

We model ecological phenomena with the help of the potentials introduced in equations (3) and (20) as a consequence of the energy functional. They are mathematical constructs that can be measured and validated only through a mathematical relation with observable quantities like densities and abundances. At best, we can claim that these mathematical constructs describe our observations appropriately. The situation is no different in physics, see Supplementary Notes.

After line 872 we added the paragraph

The potentials we introduce in equation (3) can be viewed as the Ansatz we make in DFTe. This may well prove inappropriate for certain properties of some ecosystems, but we know of no such exceptions at present. Regarding the question of existence of such potentials, we note that they cannot be measured directly. The same holds for observables in physics like electric potentials, which we have to infer by measuring the force exerted on a test charge, which itself needs to be inferred from something like a spring scale, where we read off a change of the spring's length—finally, the directly measurable quantity. Whether the electric potential 'really exists' or whether it is a mere mathematical construct is a question outside the scientific realm. However, the answer to the question of how to derive the DFTe potentials from microscopic interactions would promote the currently phenomenological DFTe to a fundamental theory.

or in eq.6: the second functional derivative of $E_\gamma[n]$ wrt $n_s(r)$, $n'_s(r')$ must be symmetric and so must the interaction $\gamma_{ss'}(r, r')$. What is the biological reason for this?

Indeed, the second functional derivatives commute: $\frac{\delta E_\gamma[n]}{\delta n_s(r) \delta n_{s'}(r')}$ is exactly equal to $\frac{\delta E_\gamma[n]}{\delta n_{s'}(r') \delta n_s(r)}$. But $\gamma_{ss'}(r, r')$ does not need to be symmetric in the species index and also not in the position variables. The values of the interaction kernels $\gamma_{ss'}(r, r')$, $\gamma_{s's}(r, r')$, $\gamma_{ss'}(r', r)$, and $\gamma_{s's}(r', r)$ in equation (6) may all be different. We added the derivation of the first and second functional derivative of $E_\gamma[n]$ after line 915:

Notes on the interaction energy. Equation (6) represents a large family of functionals, namely all possible bipartite interactions. In fact, sums of such functionals implement the more general form $\sum_{s,s'=1}^S \int_A(d\mathbf{r})(d\mathbf{r}') f(n_s(\mathbf{r})) \gamma_{ss'}(\mathbf{r}, \mathbf{r}') g(n_{s'}(\mathbf{r}'))$ of equation (6) through series expansions of the functions f and g . We impose no constraints on the interaction kernel γ within the general DFTe framework, but, of course, not all of these possible functionals can be realised or could even be endowed with biological meaning. Equation (3) requires the first functional derivative $V_i^\gamma[\mathbf{n}](\mathbf{r}) = \frac{\delta E_\gamma[\mathbf{n}]}{\delta n_i(\mathbf{r})}$ of $E_\gamma[\mathbf{n}]$, whose total variation is

$$\delta E_\gamma[\mathbf{n}] = \sum_{i=1}^S \int_A(d\mathbf{r}) \frac{\delta E_\gamma[\mathbf{n}]}{\delta n_i(\mathbf{r})} \delta n_i(\mathbf{r}) \quad (101)$$

by definition and reads

$$\delta E_\gamma[\mathbf{n}] = \sum_{i=1}^S \left(E_\gamma[\dots, n_{i-1}, n_i + \delta n_i, n_{i+1}, \dots] - E_\gamma[\mathbf{n}] \right)_{O(\delta n_i)} \quad (102)$$

$$= \int_A(d\mathbf{r})(d\mathbf{r}') \left(\sum_{i,s,s'=1}^S \left\{ \alpha_s n_s(\mathbf{r})^{\alpha_s-1} \delta_{is} \gamma_{ss'}(\mathbf{r}, \mathbf{r}') n_{s'}(\mathbf{r}')^{\beta_{s'}} \delta n_s(\mathbf{r}) \right\} + \sum_{i,s,s'=1}^S \left\{ \beta_{s'} n_s(\mathbf{r})^{\alpha_s} \delta_{is'} \gamma_{ss'}(\mathbf{r}, \mathbf{r}') n_{s'}(\mathbf{r}')^{\beta_{s'}-1} \delta n_{s'}(\mathbf{r}') \right\} \right) \quad (103)$$

$$= \sum_{i=1}^S \int_A(d\mathbf{r}) \int_A(d\mathbf{r}') \sum_{s=1}^S \left\{ \left[\alpha_i n_i(\mathbf{r})^{\alpha_i-1} \gamma_{is}(\mathbf{r}, \mathbf{r}') n_s(\mathbf{r}')^{\beta_s} \right] + \left[\alpha \leftrightarrow \beta \ \& \ \gamma_{is}(\mathbf{r}, \mathbf{r}') \leftrightarrow \gamma_{si}(\mathbf{r}', \mathbf{r}) \right] \right\} \delta n_i(\mathbf{r}). \quad (104)$$

In going from equation (102) to equation (103), we used equation (6) with $\gamma_{ss} = 0$ and $(x + \epsilon)^\alpha = x^\alpha + \alpha x^{\alpha-1} \epsilon + O(\epsilon^2)$. Using equation (101), we identify $V_i^\gamma[\mathbf{n}](\mathbf{r})$ in equation (104), which we obtained by swapping the summation indices s and s' (integration variables \mathbf{r} and \mathbf{r}') in the first (second) line of equation (103). As a consistency check and for completeness we now use equation (104) to show that

$$\frac{\delta E_\gamma[\mathbf{n}]}{\delta n_i(\mathbf{r}) \delta n_k(\mathbf{z})} = \frac{\delta E_\gamma[\mathbf{n}]}{\delta n_k(\mathbf{z}) \delta n_i(\mathbf{r})}. \quad (105)$$

The parametric dependence of $V_i^\gamma[\mathbf{n}](\mathbf{r})$ on \mathbf{r} appears both in n_i and in γ . It is therefore convenient to calculate the second functional derivative $\frac{\delta V_k^\gamma[\mathbf{n}](\mathbf{z})}{\delta n_i(\mathbf{r})}$ of $E_\gamma[\mathbf{n}]$ in analogy to equations (102)–(104) by rewriting $V_k^\gamma[\mathbf{n}](\mathbf{z})$ as

$$V_k^\gamma[\mathbf{n}](\mathbf{z}) = \int_A(d\mathbf{r})(d\mathbf{r}') \sum_{s,s'=1}^S \left\{ n_s(\mathbf{r})^{\alpha_s-1} A_{ss'}^k(\mathbf{r}, \mathbf{r}') n_{s'}(\mathbf{r}')^{\beta_{s'}} + n_s(\mathbf{r})^{\alpha_s} B_{ss'}^k(\mathbf{r}, \mathbf{r}') n_{s'}(\mathbf{r}')^{\beta_{s'}-1} \right\} \quad (106)$$

with $A_{ss'}^k(\mathbf{r}, \mathbf{r}') = \delta_{ks} \delta(\mathbf{z} - \mathbf{r}) \alpha_s \gamma_{ss'}(\mathbf{r}, \mathbf{r}')$ and $B_{ss'}^k(\mathbf{r}, \mathbf{r}') = \delta_{ks'} \delta(\mathbf{z} - \mathbf{r}') \beta_{s'} \gamma_{ss'}(\mathbf{r}, \mathbf{r}')$.

Then, it is straightforward to obtain the total variation of $V_k^\gamma[\mathbf{n}](\mathbf{z})$ as

$$\delta V_k^\gamma[\mathbf{n}](\mathbf{z}) = \sum_{i=1}^S \int_A (d\mathbf{r}) \frac{\delta V_k^\gamma[\mathbf{n}](\mathbf{z})}{\delta n_i(\mathbf{r})} \delta n_i(\mathbf{r}) \quad (107)$$

$$= \sum_{i=1}^S \int_A (d\mathbf{r}) \sum_{s=1}^S \int_A (d\mathbf{r}') \left\{ [f_{kis}^{\alpha\beta}(\mathbf{z}, \mathbf{r}, \mathbf{r}') + g_{kis}^{\alpha\beta}(\mathbf{z}, \mathbf{r}, \mathbf{r}')] + [\alpha \leftrightarrow \beta \ \& \ \gamma_{is}(\mathbf{a}, \mathbf{b}) \leftrightarrow \gamma_{si}(\mathbf{b}, \mathbf{a})] \right\} \delta n_i(\mathbf{r}), \quad (108)$$

where

$$f_{kis}^{\alpha\beta}(\mathbf{z}, \mathbf{r}, \mathbf{r}') = \delta_{ki} \delta(\mathbf{z} - \mathbf{r}) \alpha_i (\alpha_i - 1) n_i(\mathbf{z})^{\alpha_i - 2} \gamma_{is}(\mathbf{z}, \mathbf{r}') n_s(\mathbf{r}')^{\beta_s}, \quad (109)$$

$$g_{kis}^{\alpha\beta}(\mathbf{z}, \mathbf{r}, \mathbf{r}') = \delta_{ks} \delta(\mathbf{z} - \mathbf{r}') \alpha_i \beta_s n_i(\mathbf{r})^{\alpha_i - 1} \gamma_{is}(\mathbf{r}, \mathbf{r}') n_s(\mathbf{r}')^{\beta_s - 1}. \quad (110)$$

Both $f_{kis}^{\alpha\beta}(\mathbf{z}, \mathbf{r}, \mathbf{r}')$, which is proportional to $\delta_{ki} \delta(\mathbf{z} - \mathbf{r})$, and

$$\sum_{s=1}^S \int_A (d\mathbf{r}') \left\{ [g_{kis}^{\alpha\beta}(\mathbf{z}, \mathbf{r}, \mathbf{r}')] + [\alpha \leftrightarrow \beta \ \& \ \gamma_{is}(\mathbf{a}, \mathbf{b}) \leftrightarrow \gamma_{si}(\mathbf{b}, \mathbf{a})] \right\} = \alpha_i \beta_k n_i(\mathbf{z})^{\alpha_i - 1} \gamma_{ik}(\mathbf{r}, \mathbf{z}) n_k(\mathbf{z})^{\beta_k - 1} + \beta_i \alpha_k n_i(\mathbf{z})^{\beta_i - 1} \gamma_{ki}(\mathbf{z}, \mathbf{r}) n_k(\mathbf{z})^{\alpha_k - 1} \quad (111)$$

are invariant under the combined swap $[i \leftrightarrow k \ \& \ \mathbf{r} \leftrightarrow \mathbf{z}]$. Hence, the second functional derivatives of $E_\gamma[\mathbf{n}]$ commute. Of course, this also holds in the special case of $\alpha_s = \beta_s = 1$, where $\frac{\delta E_\gamma[\mathbf{n}]}{\delta n_i(\mathbf{r}) \delta n_k(\mathbf{z})} = \gamma_{ik}(\mathbf{r}, \mathbf{z}) + \gamma_{ki}(\mathbf{z}, \mathbf{r})$, which reduces further to $\frac{\delta E_\gamma[\mathbf{n}]}{\delta n_1(\mathbf{r}) \delta n_1(\mathbf{z})} = \gamma_{11}(|\mathbf{r} - \mathbf{z}|)$ if the interaction in a single-species system depends on distance alone, which is an often studied situation in physics.

We hope that we understood and addressed the Referee's concerns adequately, but would be happy to provide further clarifications if necessary.

Actually, we know that sometimes the presence of constants of motion in the dynamics may be a signature of an unrealistic ecological behaviour, like in the classical LV equations. Also, why should we expect species to engage in long-range interactions of Coulomb-type as assumed in eq.24? I perfectly understand the reasons behind those choices, which stem from classical and quantum physics. DFT has been applied to a huge number of many-body problems in physics and its success is unquestionable. In ecology, however, we neither have an underlying mathematical theory nor we measure energies as in physics, so we do not have direct control of the form of the energy functional, we do not have an underlying theory which could tell us how this energy functional should look like.

Author Comment 9. Indeed, the choice of the energy functional has to follow plausible assumptions that are informed by the researcher's insights into the specific ecological system, like long-range or at least finite-range interactions for the fruit flies, see also Author Comment 16. Here, choices other than the Coulomb-type interactions may very well prove appropriate. That is, there is a certain freedom in setting up the DFTe energy functional, which can only be reduced by constraints from additional empirical data specific to the considered ecosystem. For the fruit flies system we simply used the first long-range interaction that came to our minds, and we did not try another interaction kernel. There is likely a better choice than the Coulomb interaction, but our point is that rudimentary insights suffice to produce results on a par with existing theories, as we mention on lines 370–372.

That's why for me this is not a mechanistic/fundamental approach — as the authors claim across the paper —, but a phenomenological one, in which some well known physical principles/ properties are mapped/applied to ecology, without knowing whether this is appropriate or how good this mapping is. The energy functionals they suggest are plausible, but they do not emerge from a microscopic theory.

Author Comment 10. We noted explicitly the phenomenological nature of DFTE on lines 106–107 and claimed nowhere that DFTE is a fundamental approach or that the energy functionals emerge from a microscopic theory. However, we argue that DFTE also has a strong mechanistic component, see Author Comment 7. We agree with the Referee that we justify the mapping of the DFT methodology to ecology only by benchmarking against empirical data. Showing that this works for a wide variety of systems and delivering plausible interpretations of the DFTE ingredients—thereby establishing a novel approach to ecosystem modelling—is the purpose of our article.

More specific comments

1. I did not find eq.2 in the main text particularly illuminating. I would rather explain the terms in eq.1 as the authors did in the Methods section with eq.(4-7).

Author Comment 11.

As stated on line 102, we believe that equations (2) and (3) are the essence of DFTE in terms of interpreting the methodology, as explained on lines 102–105. We cannot remove that passage, but would be grateful for any advice the Referee could give on improving the narrative. We tried to better illuminate the importance of equation (2) by rephrasing the whole paragraph around equations (2) and (3), lines 91–101:

Among the various implementations of DFT, we find that density-potential functional theory (DPFT) [28–33] is uniquely qualified for achieving our objectives. When combined with our energy functional for dispersal (equation (4)), DPFT delivers the equilibrium density

$$n_s[V_s - \mu_s](\mathbf{r}) = \frac{1}{\tau_s} [\mu_s - V_s[\mathbf{n}](\mathbf{r})]_+$$

of species s , where the operator $[\]_+$ ensures non-negative densities by replacing negative arguments with zero, μ_s is a species-specific Lagrange multiplier that enforces the density constraint $\int_A(d\mathbf{r}) \mathbf{n}(\mathbf{r}) = \mathbf{N}$ in a focal area A (see Methods), and

$$V_s[\mathbf{n}](\mathbf{r}) = V_s^{\text{env}}(\mathbf{r}) + V_s^{\text{int}}[\mathbf{n}](\mathbf{r})$$

is the potential energy as a function of all species (density vector \mathbf{n}) at position \mathbf{r} , which merges the environment V_s^{env} (as perceived by species s ; see equation (5)) with the interaction potential $V_s^{\text{int}}[\mathbf{n}]$ (the functional derivative of $E_{\text{int}}[\mathbf{n}]$), coupling s to the other species. The simple relation between density n_s and effective potential V_s in equation (2) arises directly from our choice of functional form for the dispersal energy (equation (4)). This functional form suffices for the proof-of-principle examples studied here, although other functional forms are also plausible (e.g., equation (6) in the Methods). Henceforth, we omit arguments of functions for brevity wherever the command of clarity permits.

We agree with the Referee that the interpretations of equations (4)–(7) given in Methods would enhance the narrative in Results around equations (1)–(3). We decided to move the content of lines 391–401, which contain equations (4) and (5), but kept the more technical equations (6) and (7) in Methods, see

Author Comment 3.

We replaced the sentence on lines 124–127 by

Although this is only one of potentially many ways a computational framework for DFTE could be constructed, it provides substantial flexibility in terms of the forms of the species-interaction functionals $E_{\text{int}}[\mathbf{n}]$ (see Methods), and we will see that it describes ecological data very well.

2. the operator $[\dots]_+$ replaces negative densities with zero. This might be mathematically inconsistent, especially in the absence of a temporal dynamics, because negative densities may be the signature of instabilities. For instance, spurious effects have been found in theories such as the phase field crystal theory for matter. More safely, it is preferable that the energy functional itself ensures that densities are nonnegative.

We thank the Referee for alerting us to this issue, which is simply a typo. The operator $[x]_+ = x \eta(x)$ is in fact part of the Thomas–Fermi functional in equation (22), as derived in reference [30]— we have now corrected equation (22)

$$E_{\text{dis}}^{\text{L}}[\mathbf{V} - \boldsymbol{\mu}] = -\frac{1}{2} \sum_{s=1}^S \int (\text{d}\mathbf{r}) \frac{1}{\tau_s} [\mu_s - V_s]_+^2,$$

Alternative dispersal kernels that do not involve the Heaviside step function $\eta(\cdot)$ and guarantee nonnegative densities are well conceivable, but we do not see the need to follow that route at present.

3. I guess that multiple local minima are frequently encountered in the DFTE, so I would have expected the authors had implemented some sort of simulated annealing, but they did not mention this option. Why?

We have chosen multi-start conjugate gradient descent since it is a (quite possibly the most) popular optimiser in the DFT community, and it does the job for our present proof-of-principle work. However, we agree with the Referee that alternative stochastic optimisers such as simulated annealing or evolutionary algorithms may search for the global optimum more efficiently. While we have not prioritised a high computational efficiency, it is on our agenda for a refined DFTE methodology, and we have used particle swarm optimisation to address the high-dimensional BCI example, see Author Comment 4.

Also, to me it is not clear at all whether and on what scales the system may equilibrate. If not, it is hard to infer the correctness of density profiles which might come from metastable states and may be spurious.

Author Comment 12. The discussion of conditions under which real ecosystems actually equilibrate, is beyond the scope of our work. Regarding equilibration from the DFTe-modelling point of view and the question of metastability, we added on line 474:

There is always an equilibrium if the energy functional is bounded from below, together with the fact that the support (abundances/densities) of the energy functional is finite in any practical application. If some DFTe energy components are chosen (too) negative, the system can be unstable, in which case the energy functional has no minimum and is inappropriate for modelling the equilibrium in question. This means that another energy functional has to be considered, or, in the worst case, that DFTe is incapable of simulating this system. We also caution that no numerical optimisation algorithm for non-convex black-box functions can guarantee to find the global minimum, not even approximately. Without analytically available characteristics of the global minimum, all one may hope for are candidates of the minimiser, and those may not even be local minima—there is no way to be certain that an optimum proposed by a numerical optimisation algorithm is stable.

4. The author refer on p.10-11 to 'transient dynamics' and 'non equilibrium steady states'. This is quite odd to me, given that the current formulation is not dynamical at all. I think the authors have in mind a transient and non equilibrium steady states in a computational sense, situations which occur in the optimisation process. This is very different from an actual temporal transient dynamics and sometimes may be even spurious. At any rate, the temporal and computational scales of the phenomena may be completely unrelated. Anyway, I find at least premature such statements. Similar considerations apply to the cyclic dynamics fig.5, in which it may well be that eq.30 is a constant of motion of some LV equations, not shown here.

Author Comment 13. We agree with the Referee that the dynamics referred to in Figs. 4 and 5 are likely not realistic. In fact, they are not even real-time dynamics, as the Referee correctly states, because time is not part of the DFTe formalism for now, such that the trajectories referred to in Figs. 4 and 5 cannot be mapped unambiguously into a time series, as declared on lines 265–267. But they are not dynamics in a computational sense either—actually, we discuss on lines 1433–1456 how real-time dynamics on the dispersal time scale could potentially be extracted from the computational dynamics of the optimisation process.

On lines 271–275 in the revised manuscript, we now write

We used our case study of competing grasses to illustrate the potential for DFTe to approximate transient dynamics by assuming that the trajectory between the initial and equilibrium states is linear in the total abundance of each species. Using this assumption, we ran a simulation in which two grasses invade the equilibrium monoculture distribution of the third grass (Fig. 4).

And on lines 290–293 in the revised manuscript, we now write

Panels **b–d** also approximate time-dependent dynamics by showing snapshots of density distributions along the linear trajectory in the space of abundances from an initial state where Poa is in monoculture ($\mathbf{N} = (2.499, 0.0, 0.0)$) (see Extended Data Fig. 2a) to the global equilibrium at $\mathbf{N} = \hat{\mathbf{N}} = (1.006, 3.069, 1.738)$, as indicated by the red dots in the sketches.

Also the red equipotential line in Fig. 5 lacks information on real-time dependence, see lines 265–267, which would be necessary for comparing DFTE with time-resolved dynamical models such as the Lotka–Volterra equations, which we were not able to connect with equation (30). For clarification we replaced the sentence on lines 253–254 by

The examples of Figs. 4 and 5 are exploratory studies that demonstrate that the DFTE hypersurface can be an appropriate platform for dynamics and that DFTE can potentially be augmented with actual temporal dynamics, thereby opening the door to modelling time-dependent phenomena in ecology with DFTE (see Supplementary Notes).

See also Author Comment 18.

5. It seems to me that DFTE can be fruitfully applied when dealing with a relative small number of species, namely, small number of density fields. The scalability of the approach is unclear to me and should be addressed in more detail in the text.

Although the seven species of the synthetic food web in Fig. 6 comprise the largest number of species we had so far considered explicitly, there is no fundamental reason that would prevent investigations of many more species, see Author Comment 4 for our 20-species simulation in response to the issue of scalability in terms of number of species. We discuss the spatial scalability of DFTE on lines 372–376, 517–520, and 1375–1384.

6. Energy in eq.4 models how conspecifics repel each other in space. However, there is a large literature confirming that plant conspecifics are aggregated in space. How do you deal with that case?

We added on line 489:

The dispersal energy $E_{\text{dis}}[n_s]$ in equation (4), where an increase of the dispersal pressure constants τ_s tends to dilute n_s (see also Extended Data Fig. 1), is only one component of the total energy and has to be balanced against, for example, the environmental energy. This trade-off produces the aggregation of, for example the fruit flies in Fig. 2 and the grasses in Fig. 4. In the latter case the confining role is played by the interaction potential, not the environmental potential, which are combined anyway through equation (3). If E_{dis} with positive τ_s were the only energy component in an unconstraining, i.e., flat and infinite environment, then $\min_{n_s} E_{\text{dis}}[n_s] = 0$ for a density that is maximally dispersed/diluted with $n_s(\mathbf{r}) \rightarrow 0$ everywhere.

7. When τ_s is small in eq.4 — as in many cases — then eq.2 becomes unstable. How do you deal with this technical problem?

For negative τ_s , which would amount to conspecific aggregation instead of repulsion, the DFTE energy of equation (1) could become unstable, see Author Comment 12. For small positive or vanishing τ_s , our technical solution is given on lines 497–500: We use any τ_s that renders equation (2) technically feasible and compensate the so introduced excess energy by subtracting it from the interaction energy in equation (3). This augmentation of the selfconsistent equations (2) and (3) is a successfully tested technique.

8. As a challenge for the approach, and my curiosity, I would ask the authors to model an experiment in which bet hedging occurs. I am not sure how this could be transparently implemented in their framework.

Author Comment 14. We thank the Referee for this interesting and valuable question and believe that our answer illustrates the ease of implementing even very specific phenomena through energy functionals. After line 883, we added the paragraph

To once again illustrate the reasoning and simple intuition behind building DFTE energy components, we propose a model of the specific ecological phenomenon of bet hedging. This goes—specifically, not conceptually—beyond the ‘major ecological principles’ alluded to in the Discussion, but it does not go beyond what we proposed in equation (9). Let us suppose that a species engages in bet hedging if it gains fitness in stressful situations at the expense of fitness in favourable situations. If stressful and favourable refer to the environment (for instance, as opposed to interactions, and just to be specific here), we could perform two calculations (0) and (B), with environmental potentials $V_{\text{env}}^{(0)}$ and $V_{\text{env}}^{(B)} = \langle V_{\text{env}}^{(0)} \rangle + b (V_{\text{env}}^{(0)} - \langle V_{\text{env}}^{(0)} \rangle)$ perceived by a species without (0) and with (B) bet hedging strategy, respectively. We amplify bet hedging by decreasing the parameter b from 1 to 0: We effectively decrease large (stressful) values of $V_{\text{env}}^{(0)}$ and increase small (favourable) values of $V_{\text{env}}^{(0)}$ towards the mean $\langle V_{\text{env}}^{(0)} \rangle$. Then, $b = 0$ represents maximal bet hedging because $V_{\text{env}}^{(B)} = \langle V_{\text{env}}^{(0)} \rangle$ is constant, and the discrimination between stressful and favourable environments disappears. If, for a particular choice of b and taking into account all other constraints like resources, interactions, etc., the abundances obtained with the DFTE energy minimisation are $\hat{N}^{(B)} > \hat{N}^{(0)}$, then we may declare bet hedging ‘of relative magnitude $1 - b$ ’ advantageous. This illustrates how DFTE can incorporate, in both an abstract and a simple way, potentially very specific ecological phenomena like different egg sizes in a clutch that may have evolved as a protection against environmental variability.

and added ‘Additional illustrations of the intuition behind constructing energy functionals for ecological phenomena are provided in the Supplementary Notes.’ on lines 458–459 in the revised manuscript, which also refers to the ‘analytically solvable minimal example’ on lines 917–993, and the hypothetical parameterisation on lines 1055–1059.

9. FRUIT FLY EXPERIMENT. The qualitative behaviour is captured by the models presented, but I would not commit to one of them in particular, so I would not consider these outcomes as strong evidence in favour of DFTE.

Author Comment 15. We never meant the generalistic DFTE framework to outperform specialised approaches, as indicated on lines 129–130. We rather wanted to showcase the capacity of DFTE to deliver results on a par with established theories while adding generality, see lines 370–372.

Based on the provided information, it is difficult for me to understand why the energy on eq.24 should include a quadratic term in the environment contribution and a $1/r$ repulsive potential. Had one to analyse different experimental settings, should we include the same functional forms? Probably different exponents are preferable... This is somehow an instance of the potential drawbacks of this approach. We do not know how general are the energy functionals we write down, so one gets the impression that every system is a specific example/model that needs to be modelled in its own way. We loose the big picture and what different systems share.

Author Comment 16. We agree with the Referee (and acknowledge on lines 366–370) that one challenge in applying DFTe lies in the freedom to set up a variety of plausible energy functionals for a specific setting. We fully agree with the implied assessment that a universal, essentially parameter-free functional that, once set up, works for all specific examples, would provide a most powerful tool and would undoubtedly paint a big picture. While we fear that such a goal is unrealistic, we argue that DFTe in its current form provides a big picture nonetheless: We added on lines 413–414 in the revised manuscript:

No other single theoretical ecological framework has been shown to model such a broad range of systems.

And on line 569 we added:

By comparing, for example, the energies in equations (29) and (30) for the grasses and the microbial predator–prey system, respectively, we see that DFTe is capable of determining what (very) different systems share—in this case the parasitic interaction, but not the dispersal pressure.

Regarding the Referee’s questions about the fruit fly model in particular, we note that the entire reasoning behind the environmental and interaction energy in equation (24) is spelled out on lines 503–507, 720–722, and 728–730. We did not consider other functional forms, see also Author Comment 6 and Author Comment 9—simply, because it worked just fine with the educated guess reported here. We clarified this by adding

We acknowledge in the Discussion that the intuition required for setting up a DFTe energy functional can be a barrier of entry for applying DFTe. Such difficulties are not unexpected when introducing a fundamentally novel approach, and we can only hope that detailed accounts of our reasoning will be adequate guides. Therefore, we offer an alternative phrasing of the reasoning behind equation (24): Equation (2) is essentially $n(x) = \text{constant} - V(x)$, and the noninteracting situation $V(x) = V^{\text{env}}(x)$ produces a first approximation (cf. $n_{\tau=1}^{(0)}$ in Extended Data Fig. 1b) of $n_{\text{ref}}^{\text{Dc}}(x)$ if $V^{\text{env}}(x)$ is proportional to something like $x^{3/2}$, or x^2 , or $x^{5/2}$, which we guess just by looking at the red dashed line in Extended Data Fig. 1b; let us take x^2 . Since we implicitly assume that all flies are identical, the interaction between two flies located at x and x' has to be symmetric, i.e., the interaction integrand of equation (6) is $n(x)^\alpha \gamma(x, x') n(x')^\alpha$. The existence of territoriality among fruit flies tells us that the interaction kernel $\gamma(x, x')$ should be positive and should have a finite range. Hence, $\gamma(x, x')$ depends in some way on $|x - x'|$. Since we deem it natural to assume that γ decreases with distance, the parsimonious choice is $1/|x - x'|$. Finally, we opt for the parsimonious choice $\alpha = 1$. The dispersal energy in equation (24), which facilitates execution of the selfconsistent DPFT loop, could have been rendered irrelevant if the fit parameters ε and/or γ had turned out very large (in terms of the energies they produce relative to the dispersal energy). In a sense, the data itself takes care of suppressing irrelevant energy components.

after line 1016, and referred to this paragraph by adding ‘In the Supplementary Notes we offer a more detailed account of the reasoning behind equation (24).’ on line 509. We also added on line 730:

The dispersal energy turns out to be less than 8% of the total energy. If this is not negligible, then the fruit flies engage in a contact-type repulsion alongside the finite-range repulsion, which comprises about 32% of the total energy.

10. RESOURCE COMPETITION EXPERIMENT. It is not clear to me why the authors assume an interaction kernel as defined in eq.25, in which the exponent κ introduces implicitly a trade-off between the fitness proxies f_s and g_s . This form could not have been guessed, unless some information was already known to the modeller. If I am not missing something, the authors should acknowledge that. The approach here needs some external guidance, which does not come from the framework itself. Also, are the proxies obtained as fitted parameters? If not, how did you calculate them? The interaction energy in eq.28 should be quadratic in N , I guess, hence with a term $\sum \gamma_{ss'} N_s N_{s'}$; why is it not the case?

Author Comment 17. Indeed, our approach needs some external guidance that does not come from the framework itself, which we acknowledged on lines 366–372. The mathematical formulation of plausible energy functionals for specific settings needs to be guided by biological knowledge and intuition, see also Author Comments 9 and 16. After line 1038, we added explanations that complement our reasoning on lines 206–209 and 521–539 for setting up equation (28), including the fitness proxies, their trade-off, and the interaction energy. In this new paragraph, we incorporated lines 1039–1042:

The suspension for the diatoms represents a uniform environment with nutrient medium. That is, we may choose $V_s^{\text{env}} = 0$ for all species s , and consider the resource energy as part of the total energy. We disregard dispersal energy until we need it—our simulations eventually suggest that we do not. Hence, $E(\mathbf{N}) = E_{\text{Res}}(\mathbf{N}) + E_\gamma(\mathbf{N})$. Since Tilman’s experiments deal with competitive exclusion, we consider $E_\gamma(\mathbf{N})$ to represent amensalism and, alternatively, what we term asymmetric competition in Extended Data Tab. 1. Both interactions assign a competitive advantage of one species over another, in contrast to the symmetric repulsion of type $N_s \gamma_{ss'} N_{s'}$. We can settle for amensalism only after our simulations with the asymmetric competition yield results that contradict the data. First, however, we have to specify the interaction kernels $\gamma_{ss'}$, which we use for both amensalism and asymmetric competition, see equation (12). In the setup of Ref. [1] the algae are primarily characterised through the R^* -values

$$(R_{sk}^*) = \begin{pmatrix} 1.0 & 0.005 \\ 1.0 & 0.004 \\ 5.7 & 0.002 \\ 9.7 & 0.02 \end{pmatrix} [\mu\text{mol/L}]$$

and nutrient requirements

$$(v_{ks}) = \begin{pmatrix} 9.7 \times 10^{-7} & 1.5 \times 10^{-6} & 5.8 \times 10^{-5} & 6.3 \times 10^{-6} \\ 4.7 \times 10^{-8} & 2.6 \times 10^{-8} & 1.1 \times 10^{-7} & 1.9 \times 10^{-7} \end{pmatrix} [\mu\text{mol/cell}]$$

that are taken from Tab. 1 of Ref. [1] and enter equation (28). Lacking other options, we construct fitness proxies from these two quantities. We find it plausible that smaller R^* -values and smaller nutrient requirements could both indicate higher fitness.

That is, one parsimonious choice of fitness proxies are the inverses of these values, or rather the sum of inverses over the $K = 2$ resources. We thus arrive at the two potentially influential fitness proxies f_s and g_s of equations (26) and (27) by including the weights w_k , which acknowledge that the relative importance of resources varies depending on the supplied resource combination. We may then postulate the generic interaction kernels $\gamma_f [f_s/f_{s'} - 1]_+$ and $\gamma_g [g_s/g_{s'} - 1]_+$ according to Extended Data Tab. 1, with two fit parameters γ_f and γ_g that imply two interaction energies of the type in equation (12). The magnitudes of γ_f and γ_g inform us about the absolute and relative importance of both types of fitness proxies and their associated interaction energies. Alternatively, we can set up the single interaction kernel in equation (25), where the relative importance of both types of fitness proxies is cast into the exponent κ (where $\kappa \gg 1$ renders the fitness proxies g_s irrelevant), and where γ encodes the importance of the interaction energy relative to (and thus enables the trade-off with) the resource energy E_{Res} . The only free parameter in E_{Res} is σ . Our choice $\sigma \rightarrow -\infty$ makes nonlimiting resources irrelevant. Finally, we note that the energy function $E(N)$ in equation (28) requires input from monoculture data only.

We referred to these complementary explanations by adding on line 532:

In the Supplementary Notes we provide a complementary step-by-step account of the intuition that leads us to equations (25)–(28).

11. PLANTS IN SALINITY GRADIENTS. Fig.4 show the ability of DFTE to project data taken from uniform experiments to spatial information. This is interesting, but it is not clear how we could assess the outcomes of the equilibrium distributions in fig.4d. They seem to reproduce well the profiles in fig.4a, but this could have probably done by an envelope model as well. Do the authors get a good quantitative agreement? But how better is that wrt a null model or other approaches?

Author Comment 18. We emphasise that the distributions of Poa and Hord in Fig. 4d correlate positively with their respective limiting resource in Fig. 4a, but Pucc correlates negatively. This mirrors qualitatively the experimental data of the uniform settings reported in Ref. [36], see Supplementary Fig. 3a, and is thus not an unexpected prediction of DFTE. The correlation of Poa with its limiting resource is assessed in Supplementary Fig. 4, and the discussion in Extended Data Fig. 3 helps us to connect model input and output. However, a quantitative comparison of Fig. 4d with the results of other approaches like an envelope model is informative only to some extent, because the only data we use are those in Supplementary Fig. 3a. As declared on lines 240–241, 543–545, and 1084–1086, scarcity of data is precisely the challenge we wanted to take on with this case study. Hence, the postulated DFTE energy necessarily incorporates numerous assumptions, as spelled out on lines 547–554, and will therefore inevitably differ substantially *in content, not only in structure*, from an alternative model. That is, there is always the possibility that any discrepancies between the predictions of DFTE and those of the alternative model can be attributed to the fundamentally differing model structures. And without additional data for testing, we cannot determine which approach is superior. Nevertheless, we added panel e to Fig. 4:

FIG. 4. An application to plants in salinity gradients demonstrates how DFTe can extrapolate scarce experimental data into novel more-complex settings. **a**, We fitted the DFTe functional (equation (29) in Methods) with asymmetric repulsive contact interactions between the three grass species *Poa pratensis* (Poa), *Hordeum jubatum* (Hord), and *Puccinellia nuttalliana* (Pucc) to experimental above-ground biomass data. These data are reported in Ref. 34 for monoculture and mixture setups at spatially uniform salinity levels. We used the parameterised model to predict spatial distributions of the hypothetical, but data-informed, limiting resources ρ_{lim} for each species in a synthetic landscape with heterogeneous salinity (square area $A = 1$). **b–d**, This then allows prediction of the spatial distributions of the densities of the three species. Panels **b–d** also approximate time-dependent dynamics by showing snapshots of density distributions along the linear trajectory in the space of abundances from an initial state where Poa is in monoculture ($\mathbf{N} = (2.499, 0.0, 0.0)$) (see Extended Data Fig. 2a) to the global equilibrium at $\mathbf{N} = \hat{\mathbf{N}} = (1.006, 3.069, 1.738)$, as indicated by the red dots in the sketches. The model predicts a rich zoo of phases (see Methods and Extended Data Fig. 2), including zonation as a transient state (**c**) on the way to the smooth equilibrated mixture (**d**). The large relative deviations of DFTe densities from those of a generic envelope model ($\mathbf{N}^{\text{nm}} = (0.872, 3.056, 1.881)$), which in this case represents a null model, reveal the substantial impact of heterogeneity on the grass distributions (**e**).

On lines 258–268 in the revised manuscript, we now write

The DFTe predictions were substantially different from a generic envelope model for species distribution modelling (Fig. 4d,e and Supplementary Fig. 5), which simply maps the observed abundances in mixture in different homogeneous environments on to the heterogeneous environment and, unlike our DFTe model, fails to account for changes in species interactions in the heterogeneous environment. The envelope model’s predictions are unlikely to be accurate, because the mixture experiments of Ref. [34] already demonstrate that the species do interact in homogeneous environments (mixture abundances are not just rescaled monoculture abundances), and it is likely that these interactions would change in heterogeneous environments. We emphasise, however, that no final verdict on the accuracy of either model, DFTe or the envelope model, can be reached without additional data that test the predictions.

Note that we had implicitly ruled out the null hypothesis of ‘noninteracting grasses’ on lines 1100–1101. After line 1161, we added the new Supplementary Figure

Supplementary Fig. 5. **Grass densities from an envelope model.** The absolute densities of Poa, Hord, and Pucc obtained for the salinity landscape of Fig. 4a in the envelope model that we employed in Fig. 4e and that is equivalent to the null model of the hypothesis that ‘heterogeneous salinity has no effect on grass distributions’.

The authors highlight they get zonation as a 'non-trivial result from our DFTe simulation'. Quite the contrary, I would rather say that this is highly expected, given that they introduce by hand asymmetric repulsive contact interactions. In this case and as alluded to before, this is not an emergent behaviour coming from a microscopic understanding of the nature of the interactions between grass species, but a phenomenological modelling of the interactions as they appear at that very scale, and as implemented in the energy functional.

Author Comment 19. We did not mean emergent in the sense of emerging from a microscopic description, which we claim nowhere, see also Author Comment 7. We meant the zonation to emerge *along a trajectory* from invasion to equilibrium, which we state explicitly on lines 752–754, see also Author Comment 13, and as a result from a shifting importance of the various energy components along this trajectory, as mentioned in the caption of Extended Data Fig. 2. However, as the Referee correctly points out, the zonation *does* result from the nature of the interactions between grass species. With 'non-trivial' we merely meant to emphasise that it is not immediately obvious how to obtain discontinuities in a smooth environment.

In that regard, we would like to offer a general remark: The predictions of models are a consequence of what we include in the models in the first place. It is just a question of where and how model predictions show up. If we find a result that we had not put in right at the start (in form of the ingredients that give rise to this result), we made a computational mistake on the way. We would argue that having intuitive interpretations of model ingredients, that is, knowing to some extent what we put in, is one advantage of a mechanistic theory like DFTe. We know where to look, if we don't find what we want during model validation. And we can still be surprised by the outcomes despite the *allegedly visible* causal connections between model input and output, because even simple model structures can be too rich in possible outcomes for us to grasp at first glance. This is why we perform numerical simulations in the first place—rather than knowing the outcome from inspecting the model ingredients.

In our example of the plants in salinity gradients, we also tested asymmetric amensalistic interactions and symmetric repulsive interactions, for which we did *not* find zonation, see Supplementary Fig. 4, and which we dismissed because of conflicting qualitative field observations, as mentioned on lines 1111–1112. We added on line 1161:

Note that we could have found zonation with the alternative interactions of amensalism and repulsion since, for example, symmetric repulsive interactions in quantum gases do show spatial separations akin to zonation [Trappe, M.-I. *et al.* Phase transitions of repulsive two-component Fermi gases in two dimensions, *New J. Phys.* **23**, 103042 (2021)].

We hope that the Referee will find the above revisions adequate—together with our amendments in response to the editor's and to the other referees' comments.

Response to Referee 3:

All page and line numbers, references, and equations mentioned in the authors' responses refer to the original manuscript (submitted on June 16, 2021), unless stated otherwise. Grey-shaded boxes are reviewer comments. Unshaded boxes with black frame are our amendments of the original manuscript. Text outside of boxes is directed at the reviewer. Some of our responses are in boxes with green frame, labelled with 'Author Comment' (clickable), and referred to in other responses, see page 1 for an overview.

Reviewer 3 (Remarks to the Author):

The authors introduce a standard method of many-body physics (density functional theory) to study multi-species communities. After introducing the framework, the authors apply it successfully to 5 examples, modeling synthetic or empirical data.

I believe that there is an element of interest and value in the paper. Unfortunately, it does not emerge with clarity from the text. Here below a series of major and minor points.

We thank the Referee for the kind assessment that our work delivers insights of 'interest and value' and are grateful for the Referee's advice to improve our narrative substantially—we hope that the amendments of our manuscript and our accompanying responses to the remarks of all three Referees are to the point.

We completely overhauled the narrative and phrasing around the four case studies illustrated in Figs. 2–5, on lines 132–254 (lines 175–311 in the revised manuscript), where key aspects of DFTe are discussed. We also completely overhauled the Discussion on lines 333–388 (lines 406–483 in the revised manuscript). All other amendments to our original manuscript are spelled out explicitly in the present document.

Major point

1. My main concerns regard the role of DFT in being a tool that helps the modeling, the interpretation, and/or the conceptualization of ecological data. The output of DFT is equation 2, which is a self-consistent equation for the density of $n(r)$ as a function of the interactions with other species and the environment. DFT allows to obtain eq 2 from the general minimization of eq 9. A radical difference between many-body physics and ecology is that, in the former, the Hamiltonian is known from first principle. The problem in many-body physics is then mainly computational: given a Hamiltonian of a system compute the observables. DFT allows to reduce the many-body problem into an effective single-body problem (e.g. for the electron density) from which is possible to compute the interesting observables. In Ecology we do not have the 'well-accepted' first principles from which we can write equations that describe the community dynamics (letting aside the fact that the equations that we typically write in ecology are not Hamiltonian). If we map the fixed points of the community dynamics into the minima of an effective (energy) function, there are not a-priori way to fix the form of the energy function. Contrary to many-body physics, the uncertainty in ecology does not come from the limited computational power that we have to solve the equations of motion, but from the uncertainty that we have around the equation of motions themselves.
2. The motivation behind the particular choice of the functional forms of the energy functions is unjustified.

Author Comment 20. We fully agree with the Referee that physics-type Hamiltonians, viz. energy function(al)s, are yet unknown in ecology, and that mapping one point (the minimum) onto an effective function (that is to be minimised) is an entirely ambiguous enterprise—until the predictions implied by the postulated function are validated against new data. We reduced the ambiguity in setting up the

energy functional to some degree by invoking the principle of parsimony, which we had not mentioned explicitly in the manuscript, see also Author Comment 16. Therefore, we now write on lines 456–471 in the revised manuscript:

With our range of examples (summarised in Fig. 1), we have highlighted that rudimentary insight into ecosystem functioning can be sufficient for generating results on par with established approaches. Additional illustrations of the intuition behind constructing energy functionals for ecological phenomena are provided in the Supplementary Notes. We judge the main technical challenge in future applications of DFTE to be the parameterisation of models at landscape scales relevant to many ecological applications. Such applications will require not only more data and careful choice of parsimonious functional forms for the energy components, but also high-performance computing.

We have demonstrated the potential for DFTE to provide a unified modelling framework for ecology via applications to a variety of systems, from predatory microbes in test tubes to interacting grasses in saline landscapes to a synthetic food web. Our approach is inspired by DFT in physics, with the key difference that, whereas the DFT energy functional is derived from first principles, the DFTE energy functional must be established via ecological intuition and parsimony. Ultimately, as with any ecological model, the arbiter of the value of DFTE must be determined by its ability to fit and predict ecological data, and on this count our applications have shown DFTE to be comparable to existing specialised approaches.

We also replaced the sentence on lines 87–90 by

Rather, we use our DFTE and its energy functional E , derived by analogy with DFT and informed by ecological principles, as a tool for parameterising sub-components of ecological systems and then combining the components to generate higher-order predictions. Furthermore, in ecology our data of interest are species abundances, resource levels and related variables, rather than energy. To the extent that the hypothesised mathematical structure of E is correct in any given instance, the predictions of these ecological variables will be accurate.

Until the functional forms can be derived as emerging from microscopic first-principles interactions, DFTE is a partially phenomenological theory, see Author Comment 7. The justification of the functional forms comes solely from the successful benchmarking against empirical data. In other words, we let the data tell us, though ambiguously to some extent, what the functional form is, see also Author Comment 6.

For instance, in equation 30, the interaction between *P. aurelia* and *D. nasutum* is modeled as a sum of quadratic forms, plus a term proportional to $N_D N_P^2$. Instead of raising the first two terms to the second power, one could have taken any convex function with a minimum in zero. Whether these alternatives, yet a-priori equivalent in terms of assumptions, choices would give the same results is unclear.

Equations (28) and (30) represent two systems with resource dependence, where we can (i) establish and (ii) validate the form of the resource energy functional through empirical data. We chose the quadratic form of the resource energy for its simplicity and were able to match the DFTE predator-prey cycle to the data with reasonable accuracy. Although there is no doubt that resource energies other than quadratic forms can produce similar results, see also Author Comment 9, sufficiently strong deviations from the quadratic form will render the predictions less accurate. While we did not try other resource functionals, we replaced the common-sense choice of a parasitic interaction by amensalism, which leads to far less

accurate DFTe predictions and which, in turn, makes a point in favour of the quadratic form of the resource energy that shapes the DFTe hypersurface in conjunction with the interaction, see lines 781–787. We may argue that the consistency of equation (28) with the algae data establishes the quadratic form of the resource energy, and the successful predictions based on equation (30) validate it. Of course, two examples do not suffice for declaring a functional universally applicable. We hope that the majority of ecosystems can be modelled with such simple functional forms, and we have not found a counterexample yet, but we surmise that some systems will require different functional forms, see Author Comment 6.

3. Related to the previous point, the usage of the word 'mechanistic' is quite unclear in this context. The specific form of the Hamiltonian is chosen to be a sum of convex functions each with a minimum in the 'right' position. There is no 'mechanistic' interpretation of the shape of these functions. I am not arguing against 'effective' models (any model is effective), but I am wondering of what the authors mean with 'mechanistic' in the title.

We thank the Referee for compelling us to better explain the general attributes of DFTe. We believe that Author Comment 7 and Author Comment 10 fill this gap.

4. The manuscript is very hard to follow. Here a few examples: - the narrative of the introduction and first part of results is built around the parallel with how DFT is used in many-body physics, which make many part confusing for the audience without a background in physics (e.g. 'the quantum-mechanical spin-statistics theorem' and the comparison between fermions and dispersal). - other analogies are quite hard to parse. For instance: 'quantum-gas-inspired mechanistic energy functional' is used when eq 1 is introduced. In what sense is 'quantum-gas-inspired'?

Author Comment 21. As mentioned above, we hope that the rephrasing of large parts of our manuscript makes it more accessible. Our inspiration in developing DFTe came partially from the behaviour of quantum gases, where, for example, different (atomic) species segregate or aggregate in space depending on the species-specific environments and interactions—a one-to-one paraphrasing of the situation in ecology. But we realise now that these references to quantum mechanics confuse more than they clarify. We made according amendments and revised our narrative throughout the manuscript, see Author Comments 2, 3, and 8.

In what sense 'mechanistic'? What is an 'energy functional' in ecology? - in eq 2, it appears $n_s[V_s - \mu_s](r)$. It is not explained what $[\]$ in this context? n_s is a functional of $V_s - \mu_s$ (I guess). But this is not explained when the equation appears.

We explained on pages 3 and 4 what an energy, and by implication, an energy functional, means in DFTe: It is simply a cost function. See Author Comment 3 for the revised explanations and the revised Discussion for our notion of 'mechanistic energy functional'.

To define our notation of functionals, we added

When using square brackets in $f[n]$, we declare that a function f is a functional of the function n .

on line 66. The general notation in equation (2), which declares $n_s[V_s - \mu_s](r)$ a functional is appropriate since all functions can be viewed as functionals and vice versa.

For further clarification, we added on line 500

The notation $n_s[V_s - \mu_s](\mathbf{r})$ declares that n_s is a functional (function of functions) of $V_s - \mu_s$, and that n_s is also a function of \mathbf{r} . In the specific case of equation (2), the functional dependence of n_s reduces to a trivial dependence on $V_s - \mu_s$, which renders n_s dependent on \mathbf{r} alone—a consequence of equation (22) as the source of equation (2). Alternatives to equation (22) with nonlocal integrands will make n_s a nontrivial functional of $V_s - \mu_s$, for example, $n_s[V_s - \mu_s](\mathbf{r}) = \frac{1}{\tau_s} \int (d\mathbf{r}') \frac{[\mu_s - V_s(\mathbf{r}')]_+}{|\mathbf{r} - \mathbf{r}'|}$, which reduces to equation (2) if $1/|\mathbf{r} - \mathbf{r}'|$ is replaced by $\delta(\mathbf{r} - \mathbf{r}')$.

Minor points

- 'The resolution of such questions ... temporal and spatial scales', where 'such questions' refer to the emergence of macro/mesoscopic properties from microscopic interactions. A philosophical comment / opinion, likely irrelevant. In stat mech, the emergence of the macroscopic description of reality (thermodynamics) is possible exactly because one restricts the analysis to a precise temporal (and spatial) scale. Thermodynamics work on the timescales longer than the equilibration times and shorter than whatever will impact the system in the future. And the general lesson of stat mech is that (at least close to a critical point, at least for critical exponent, which however are the main non-trivial macroscopic properties) microscopic details simply do not matter.

Author Comment 22. We agree with the Referee that statistical theories work since, roughly speaking, the microscopic details do not matter for large enough systems observed on an appropriate time scale. We claim nowhere, however, and in particular not in the abstract, that we model macro/mesoscopic properties as emerging from microscopic interactions, see also Author Comment 7.

- line 69-71. It is unclear why (and in what sense) MaxEnt is equivalent to energy minimization.

Energy minimisation in DFTE and entropy maximisation in MaxEnt share that they both invoke axiomatically that the equilibrium can be found by optimising an aggregate function of the system configurations. However, since the notion of energy we introduce in DFTE is not an established notion of energy in ecology, we removed the somewhat misleading and superfluous sentence on lines 69–71.

- the description around eq 2 and 3 is quite obscure and does not clarify where the equation comes from and what V is. It is clarified in the methods, but it would be better to make it clear also in the text.

Author Comment 23. We moved parts of the explanations from Methods and improved the narrative around equations (2) and (3), see Author Comments 2, 3, and 8.

- lines 349-351. In what sense the conceptual foundation is 'intuitive and straightforward'? How does it 'unify the major ecological principles in a single mechanistic framework'?

We acknowledge that 'intuitive' and 'straightforward' are subjective terms and removed the latter. We think, however, that the modular character of the energy functional, the simplicity of those modules, and the only rudimentary ecological insights required in their construction, are features that allow also researchers outside ecology to gain an intuitive access to ecosystem modelling. We revised the whole Discussion, including lines 343–363, see also Author Comments 6 and 7. In particular, we clarified what we meant by 'unify', see Author Comment 6. We also replaced 'straightforward' on line 881 by 'intuitive'. After line 883, we added another illustration of the intuition behind building energy functionals in DFTE—in this case for the ecological phenomenon of bet hedging, see Author Comment 14.

We also replaced ‘Our DFTE bridges ecological scales in an intuitively accessible, comprehensive framework and successfully predicts ecological data from systems with a variety of interacting taxa and environmental constraints.’ on lines 52–55 by

Our DFTE bridges ecological scales by allowing sub-components of an ecological model to be parameterised separately and then combined to predict the behaviour of more-complex systems. We first describe the general approach and then present applications to a variety of ecological systems with different properties.

- lines 353-355 + 360-361. Higher-order interactions (lines 360-361) is an example of where the components of the DFTE functional *cannot* be inferred from simpler subsystems.

On line 412 we added

Multi-partite interactions based on bipartite interactions do not seem to be an uncommon scenario [45]. However, there may be systems where nonzero coefficients $\gamma_{ss's''}$ couple all species. This poses a challenge for mechanistic theories in general. Then, ‘simpler subsystems’ that have to be included in the DFTE workflow of Fig. 1a can only refer to situations where other energy components are absent, such as resource terms or complex environments. For example, the coefficients $\gamma_{ss's''}$ could be extracted in an experiment with a controlled simple environment and then used to model the interacting species in a real-world setting.

and removed ‘, cf. Ref. [45],’ on line 411. Note also that we elaborate further on higher-order interactions on lines 410–412. Our fruit fly study presents a similar situation: The interactions and environment have to be considered together, and the ‘simple subsystems’ used for fitting refer to the simpler environment of the quasi-1D chamber and the simpler setup of only three (uncrowded, viz., barely interacting, see lines 512–514) flies in the staircase chamber. The results are used to model the crowded situation in the more complex environment of the staircase chamber.

And it’s unclear to me what is the difference with standard ecological theories (lines 351-353). If I believe in Lotka-Volterra equations, I can parameterize the equations using pairwise experiments and try to infer the outcome of the dynamics in larger communities. If it fails, one can change the model or include higher-order terms. Same for DFTE.

The scaling-up of such purely microscopic theories presents enormous difficulties in practice and has not proved successful to date, which we allude to on lines 203–205. In this context, we find it instructive that D. Tilman published his groundbreaking work on resource competition four decades ago, and yet no-one has apparently successfully extended this to systems of three or more species. This is one reason why we argue on lines 21–30 that alternative mechanistic approaches like DFTE should be developed.

We hope that the Referee will find the above revisions adequate—together with our amendments in response to the editor’s and to the other referees’ comments.

Additional amendments:

- We removed ‘and facilitates quantitative assessment of interventions’ on lines 19–20.
- We replaced ‘(see Methods)’ on line 66 by ‘(the three terms on the right-hand side of equation (1); see Methods)’, and ‘empirical and synthetic data’ on lines 110–111 by ‘empirical data and existing modelling approaches’.
- We replaced ‘theory’ on line 114 by ‘framework’, ‘diverse’ on line 125 by ‘varied’, and the sentence on lines 129–130 by ‘The DFTe framework’s strength is that it brings generality to ecosystem modelling, though it may be outperformed by specialised modelling approaches in specific cases.’
- We replaced ‘steady-state dynamics’ on lines 257–258 by ‘non-equilibrium steady-state dynamics’, ‘Finally’ on line 272 by ‘Next’, and ‘that are reminiscent of, for example,’ on line 274 by ‘intended to represent’.
- We moved the content of lines 275–286 in Results to line 576 in Methods, on line 290 after ‘hypothetical species’ we added ‘on a synthetic landscape with heterogeneous environmental suitability and resource availability (Extended Data Tab. 3 and Extended Data Fig. 4)’, and on line 301 we replaced ‘Extended Data Fig. 6’ by ‘Extended Data Figs. 5 and 6’.
- We replaced ‘realistic’ on line 308 by ‘complex’, and ‘summing (4)–(7) and (optionally)’ on lines 453–454 by ‘summing equations (2), (3), (6), and (7) and by (optionally)’.
- We added to the references the following publications, which are pertinent to our amendments:
Condit2019, <https://doi.org/10.15146/5xcp-0d46>, Ref. [42] in the revised manuscript
Davies2021, <https://doi.org/10.1016/j.biocon.2020.108907>, Ref. [43] in the revised manuscript
Wiegand2021, <https://doi.org/10.1038/s41559-021-01440-0>, Ref. [44] in the revised manuscript
Hubbell1986, Ref. [45] in the revised manuscript
Volkov2009, <https://doi.org/doi:10.1073/pnas.0903244106>, Ref. [46] in the revised manuscript
Wiegand2012, <https://doi.org/10.1098/rspb.2012.0376>, Ref. [47] in the revised manuscript
Simini2012, <https://doi.org/10.1038/nature10856>, Ref. [51] in the revised manuscript
Trappe2021, <https://doi.org/10.1088/1367-2630/ac2b51>, Ref. [57] in the revised manuscript
Mohammadi2017, <https://doi.org/10.1109/SMC.2017.8122921>, Ref. [58] in the revised manuscript
- We included Condit2019 in ‘Refs. [1, 25, 36, 50]’ on line 584 and added ‘The data for Barro Colorado Island is accessible through Dryad at <https://doi.org/10.15146/5xcp-0d46>.’ on line 591.
- We removed ‘of the uniform systems’ on line 595.
- We updated the links in the Code Availability statement on page 25, which now include the updates on Fig. 2, Fig. 4, and the new case study of Barro Colorado Island.
- We replaced equation (S13) by the more parsimonious expression $V_s^{\text{env}}(\mathbf{r}) = \sum_{k=1}^K \exp[\chi(\rho_{sk}^* - \rho_k(\mathbf{r}))]$, ‘Supplementary equation (S17)’ on line 1207 by ‘Supplementary equations (S17) and (S18)’, and line 1491 by ‘The BCI map in Fig. 1 as well as the silhouettes in Fig. 1 and in Supplementary Fig. 7 come with public domain licence: • <https://nsf.gov/news/mmg/media/images/map4.jpg> (Courtesy: National Science Foundation)’.
- We replaced ‘<https://publicdomainvectors.org/en/free-clipart/Vector-clip-art-of-bear-from-the-Flag-of-California/17297.html>’ on lines 1493–1494 by ‘<https://publicdomainvectors.org/en/free-clipart/Bunch-of-grass-vector-illustration/25652.html>’
- We removed all italicisations that were meant to convey emphasis.

Reviewer comments, second round –

Reviewer #1 (Remarks to the Author):

In this revised version, the authors have addressed in full detail all my original comments including a better description of the method, including higher levels of biodiversity, and giving a closer connection between the equations used and ecological concepts (Figure 1). Although I do not have additional comments, I still think the text is difficult to follow, but this is my personal opinion. Moreover, and considering that the journal it is multidisciplinary one, a suggestion to make it more oriented to ecologists might not be a good recommendation. Therefore, I assume the level of abstraction included here might be enough to attract to at least theoreticians, and ecologists with a strong theoretical background.

Reviewer #2 (Remarks to the Author):

[See attached review]

Honestly, after such a long time since my first review of this paper, I did not remember the details and could not recollect what I wrote about it. In addition, in this second round of revision, I was given 80 pages of main text plus supplementary information and other nearly 40 pages of rebuttal. I have not got enough time to revise all this material, but I anyway read the main text afresh.

I think the authors have done a great job in revising the paper and now it reads well with much less jargon, showing clearly an effort for connecting areas which are far apart. I still have doubts about the real power of the DFTe: in my standpoint, the problem of scalability of the approach is still open (10 or 20 species is not a big number for a landscape ecosystem). More generally, the summation of convex functions with minima in the right position does not mean that we have got the correct density functional. Therefore, this approach is fundamentally phenomenological though driven by some ecological intuition. I cannot provide further details as I did for my first round of review (3 pages), but I am somehow convinced that the framework can at least produce results on par with established frameworks. The authors show (and claim) that the DFTe is more than a merely statistical approach, though not fully mechanistic. I would still strongly suggest to better explain eq.(4) in plain words; I find the current notation somehow not easy to follow. In summary, I think this paper has the potential to stimulate further interesting discussions in theoretical ecology.

Response to Referee 1:

In this revised version, the authors have addressed in full detail all my original comments including a better description of the method, including higher levels of biodiversity, and giving a closer connection between the equations used and ecological concepts (Figure 1). Although I do not have additional comments, I still think the text is difficult to follow, but this is my personal opinion. Moreover, and considering that the journal it is multi-disciplinary one, a suggestion to make it more oriented to ecologists might not be a good recommendation. Therefore, I assume the level of abstraction included here might be enough to attract to at least theoreticians, and ecologists with a strong theoretical background.

We thank the Referee for the kind assessment of our revised manuscript and are happy to hear that our revisions resolved all the Referee's concerns.

Response to Referee 2:

Honestly, after such a long time since my first review of this paper, I did not remember the details and could not recollect what I wrote about it. In addition, in this second round of revision, I was given 80 pages of main text plus supplementary information and other nearly 40 pages of rebuttal. I have not got enough time to revise all this material, but I anyway read the main text afresh.

I think the authors have done a great job in revising the paper and now it reads well with much less jargon, showing clearly an effort for connecting areas which are far apart.

We thank the Referee for the kind assessment of our revised manuscript and hope that we have adequately addressed the remaining issues as detailed below.

I still have doubts about the real power of the DFTe: in my standpoint, the problem of scalability of the approach is still open (10 or 20 species is not a big number for a landscape ecosystem).

As alluded to on lines 460–463, we agree with the Referee that comprehensive DFTe models of ecosystems at landscape scales will likely require considering many more than 20 species of diverse taxa. Although possible in principle, practical implementations of such systems are beyond the scope of the present article.

More generally, the summation of convex functions with minima in the right position does not mean that we have got the correct density functional. Therefore, this approach is fundamentally phenomenological though driven by some ecological intuition.

We agree with the referee and acknowledged these characteristics of DFTE in the Discussion, in particular on lines 439–444, 452–459, and 466–471. Furthermore, we removed ‘mechanistic’ as an attribute of fDFTE throughout the manuscript, [redacted] we kept lines 432–448, where this attribute is contextualised and discussed.

I cannot provide further details as I did for my first round of review (3 pages), but I am somehow convinced that the framework can at least produce results on par with established frameworks. The authors show (and claim) that the DFTE is more than a merely statistical approach, though not fully mechanistic. I would still strongly suggest to better explain eq.(4) in plain words; I find the current notation somehow not easy to follow.

We extended our explanations of equation (4) in the paragraph on lines 133–139 as follows:

The coupled equations (4) and (5) are the essence of DPFT in general and of DFTE in particular. According to equation (4), the members of species s seek to reside where the cost V_s due to environment and interactions is small: A reduction of V_s increases the density, that is, we interpret small values of V_s as favourable to species s . The spatially varying values of V_s themselves are determined through the self-consistent solution of equations (4) and (5). This automatically produces a trade-off between intra-specific dispersal pressure, which rises with larger density (see Methods), and a confining potential V_s that can be interpreted as an effective environment, where interactions modify the bare environment V_s^{env} . Although this is only one of potentially many ways a computational framework for DFTE could be constructed, it provides substantial flexibility in terms of the forms of the species-interaction functionals $E_{\text{int}}[\mathbf{n}]$ (see Methods), and we will see that it describes ecological data very well.

In summary, I think this paper has the potential to stimulate further interesting discussions in theoretical ecology.

We sincerely thank the Referee for the rigorous review of our work and hope that the Referee will find our amendments adequate.

Additional amendments:

- [redacted]
- We corrected the following typos: ‘Type’ → ‘type’ on line 324; ‘ \mathbf{R} ’ → ‘ R_k ’ in equation (13); ‘Grass’ → ‘Fungus’ on line 1650.